# SLAT: Segment-Level Adaptive Trimming for Efficient CoT Reasoning

**Jian Yao** [1]  **Xiongcai Luo** [2]  **Ran Cheng** [1 3 4]  **Kay Chen Tan** [1]

## Abstract

Recent advances in Large Reasoning Models have significantly improved chain-of-thought (CoT) capabilities via reinforcement learning (RL). However, generated reasoning chains frequently suffer from structural redundancy (i.e., *overthinking*), incurring high computational overhead without improving answer correctness. Existing mitigation strategies typically rely on token-uniform length penalties, which provide coarse, segment-agnostic pressure toward shorter outputs and can inadvertently suppress useful reasoning alongside redundancy. To address this, we demonstrate that inefficiency concentrates in high-probability segments with low marginal utility. We derive a theoretical characterization of segment suboptimality under the correctness-length trade-off objective and propose SLAT (Segment-Level Adaptive Trimming), an RL framework that selectively suppresses redundant segments based on this criterion. Empirical results on standard benchmarks indicate that SLAT establishes a superior accuracy-efficiency Pareto frontier, reducing reasoning length by $50\%$ relative to uncompressed baselines while maintaining competitive accuracy. Overall, our results suggest that theoretically grounded, segment-aware trimming is a promising direction for efficient CoT reasoning in large language models.

## 1. Introduction

Recent advancements in large reasoning models (LRMs) (OpenAI, 2024; Guo et al., 2025; Team et al., 2025a;b; Zeng et al., 2025a; Seed et al., 2025), such as OpenAI-o1

---

[1]Department of Data Science and Artificial Intelligence, The Hong Kong Polytechnic University [2]Bytedance [3]The Hong Kong Polytechnic University-Daya Bay Technology and Innovation Research Institute, Huizhou,Guangdong Province, China. [4]The Hong Kong Polytechnic University Shenzhen Research Institute, Shenzhen, China. Correspondence to: Ran Cheng <ran-peter.cheng@polyu.edu.hk>.

*Proceedings of the 43rd International Conference on Machine Learning*, Seoul, South Korea. PMLR 306, 2026. Copyright 2026 by the author(s).

and DeepSeek-R1, have demonstrated significant improvements in complex reasoning tasks through reinforcement learning (RL), inspiring a growing line of follow-up work on RL-trained reasoning models (Zeng et al., 2025b; Liu et al., 2025b; Yu et al., 2025; Zhang et al., 2025e; Yao et al., 2025; Zhang et al., 2025c;a). These models refine long chain-of-thought (CoT) reasoning, leading to improved accuracy and robustness across mathematics, logic, and programming domains. However, increasing reasoning length brings substantial computational overhead and redundancy, a phenomenon often referred to as overthinking (Chen et al., 2024). This has sparked rising interest in efficient reasoning, which aims to shorten reasoning traces while maintaining strong task performance (Sui et al., 2025; Feng et al., 2025).

One way to pursue efficient reasoning is to intervene only at inference time, for example, by stopping generation once confidence stabilizes or a first coherent solution appears (Laaouach, 2025; Wei et al., 2025). Another way is to train the model so that efficiency is built into the policy, via RL or supervised fine-tuning (SFT) followed by RL (Hou et al., 2025; Luo et al., 2025; Aggarwal & Welleck, 2025; Liu et al., 2025a; Wu et al., 2025; He et al., 2025). In this line of work, a common approach is to add an explicit length penalty to discourage long CoTs (Hou et al., 2025; Aggarwal & Welleck, 2025; Liu et al., 2025a; Arora & Zanette, 2025). However, token-level length costs are coarse-grained, since they penalize all tokens equally and cannot distinguish necessary reasoning from redundant exploration. When combined with correctness rewards, they may suppress essential intermediate steps and shrink the exploration space, leading to shorter but less capable policies. This motivates a more targeted mechanism for reducing redundancy.

To better understand how such redundancy manifests in practice, we inspect individual reasoning traces of a trained reasoning model. Figure 1 shows token-level generation probabilities for the prompt "What number is a multiple of every integer?". Although the final answer is correct, the model produces a disproportionately long explanation. Importantly, once the key idea is established, the remaining CoT largely consists of restating definitions and spelling out obvious intermediate deductions rather than introducing new reasoning. A salient symptom is the repeated occurrence of the low-information phrase *"multiple of every integer"*. This phrase is echoed across successive sentences with minimal

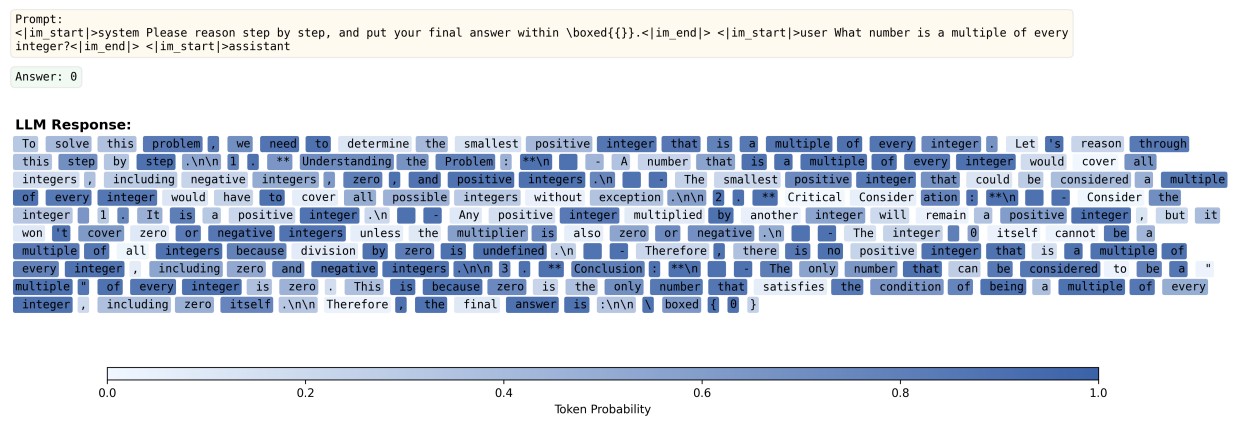

*Figure 1.* Case study of redundancy in a reasoning trace. Although the model outputs the correct answer, it generates a long CoT that contains repetitive restatements (of the problem condition and trivial deductions); these tokens form high-probability segments (darker shading), indicating near-deterministic, low-information content. Additional cases are provided in Appendix D.1.

semantic progress, forming a contiguous high-probability block in the heatmap. There are also two additional representative overthinking cases provided in Appendix D.1 (*repetition of problem statements* and *verification-style tail*), which similarly exhibit long contiguous high-probability segments with limited incremental reasoning content. Motivated by this pattern, we take a segment-level perspective and hypothesize that *inefficiency often concentrates in high-probability reasoning segments*.

This hypothesis is also consistent with recent entropy-based analyses of CoT redundancy, which argue that low-entropy generation tends to carry limited information about the final answer (Li et al., 2025). Since high-probability token runs correspond to near-deterministic, low-entropy behavior in practice, these results provide complementary support for focusing on high-probability segments as a major source of inefficiency. To make our hypothesis precise, we adopt a finer-grained, segment-level perspective and analyze it under an explicit correctness-length trade-off. Specifically, we study an objective that balances accuracy and reasoning length, and derive a sufficient condition under which generating a particular segment is suboptimal. The condition depends on both the segment length and its generation probability, and becomes easier to satisfy when a segment is long and produced with high probability. This theoretical characterization provides a principled inductive bias for efficient reasoning and motivates a trimming-based training framework that selectively suppresses such segments, encouraging concise yet faithful CoTs.

To operationalize this inductive bias in practice, we build on the Group Relative Policy Optimization (GRPO) algorithm (Shao et al., 2024) and implement it as segment-level reward shaping. For each generated answer, we run a sliding window over the reasoning trace, compute the joint probability of the tokens within each window, and assign a penalty whenever this probability exceeds a predefined threshold. The accumulated penalty is then added as an extra reward term on top of the original GRPO objective. We evaluate our method on several standard reasoning benchmarks. Empirical results show that SLAT consistently achieves a better accuracy-length trade-off than uniform length objectives, reducing the average reasoning length by around 50% relative to the corresponding uncompressed baseline while maintaining competitive accuracy. These results suggest that theory-informed, segment-level reward shaping is an effective mechanism for enhancing the reasoning efficiency of LLMs.

Our contributions are:

1. We identify high-probability segments as a common source of inefficiency in long CoT reasoning and provide a theoretical characterization of when generating such segments is suboptimal under a correctness-length trade-off, introducing a principled inductive bias for efficient reasoning.

2. Guided by this insight, we propose a trimming-based framework integrated into RL training that selectively suppresses redundant high-probability segments, encouraging concise yet faithful reasoning.

3. We validate our approach on multiple standard reasoning benchmarks, achieving substantial reductions in reasoning length while maintaining accuracy and consistently outperforming length-regularized baselines.

## 2. Preliminary

### 2.1. RL for LLMs

We cast the LLM generation pipeline as an RL problem. Here, the LLM is treated as a policy that produces outputs (actions) conditioned on input prompts (states) and receives evaluative feedback (rewards) for its generated responses. This view aligns the sequential nature of language generation with RL's state–action–reward formalism, enabling systematic behavior optimization via reward signals.

Formally, given a prompt $x \in \mathcal{X}$, we define the action space $\mathcal{O}$ as the set of potential output sequences $o = (o^1, \ldots, o^T)$ ($T$ is the length of the output). A policy $\pi_\theta(\cdot \mid x)$, parameterized by $\theta$, generates outputs conditioned on $x$ according to the distribution:

$$\pi_\theta(o|x) := \prod_t \pi_\theta(o^t|x, o^{<t}), \tag{1}$$

where $o^{<t} = (o^1, o^2, ..., o^{t-1})$.

### 2.2. GRPO algorithm

The R1-zero training method proposed by DeepSeek-R1 (Guo et al., 2025) has attracted significant research attention due to its computational efficiency and effectiveness. In our work, we adopt this training method as our backbone. R1-zero centers on the GRPO algorithm (Shao et al., 2024), which streamlines the process by eliminating the need for a separate critic model, which is usually as large as the policy model, and instead estimates baselines using group scores. Specifically, for each question $x$, GRPO samples a group of outputs $\{o_1, o_2, ..., o_G\}$ from the old policy $\pi_{\text{old}}$ and optimizes the policy $\pi_\theta$ by maximizing the following objective:

$$\mathcal{J}_0(\theta) = \mathbb{E}_{\substack{x \sim \mathcal{X}, \\ \{o_i\}_{i=1}^G \sim \pi_{\text{old}}(\cdot|x)}} \left[ \frac{1}{G} \sum_{i=1}^G \left( \min \left( \frac{\pi_\theta(o_i|x)}{\pi_{\text{old}}(o_i|x)} A_i, \right. \right. \right.$$

$$\left. \left. \left. \text{clip} \left( \frac{\pi_\theta(o_i|x)}{\pi_{\text{old}}(o_i|x)}, 1 - \epsilon, 1 + \epsilon \right) A_i \right) - \beta \, \mathbb{D}_{\text{KL}} \big( \pi_\theta \, \| \, \pi_{\text{ref}} \big) \right) \right], \tag{2}$$

where $\epsilon$ and $\beta$ are hyperparameters, the KL term is defined as

$$\mathbb{D}_{KL}(\pi_\theta||\pi_{ref}) = \frac{\pi_{ref}(o_i|x)}{\pi_\theta(o_i|x)} - \log \frac{\pi_{ref}(o_i|x)}{\pi_\theta(o_i|x)} - 1, \tag{3}$$

and the advantage $A_i$ is computed using a group of rewards $\{r_1, r_2, ..., r_G\}$:

$$A_i = \frac{r_i - \text{mean}(\{r_1, r_2, ..., r_G\})}{\text{std}(\{r_1, r_2, ..., r_G\})}. \tag{4}$$

For clarity, we denote the binary correctness reward used in GRPO as $r_{\text{ori}}$, to distinguish it from the additional reward terms introduced later.

### 2.3. CoT Reasoning

In CoT reasoning, we decompose the model output token sequence $o$ into a reasoning trace and a final answer, i.e., $o = [z \, \| \, a]$, where $z$ is the CoT token sequence generated before producing the final answer $a$. Let $\mathcal{A}$ denote the set of correct answers for $x$. We define the probability that the LLM outputs a correct answer (from the perspective of the CoT generation process) as

$$P_\theta(\mathcal{A} \mid x) = \mathbb{E}_{z \sim \pi_\theta(\cdot|x)} \, P_\theta(\mathcal{A} \mid x, z), \tag{5}$$

where $P_\theta(\mathcal{A} \mid x, z) = \sum_{a \in \mathcal{A}} \pi_\theta(a \mid x, z)$ denotes the conditional correctness probability given a specific reasoning trajectory $z$.

## 3. Method

This section develops a segment-level perspective on efficient CoT reasoning. We first formalize a correctness-length objective and analyze how a candidate reasoning segment affects this trade-off, deriving a sufficient condition under which removing that segment improves the objective. This characterization suggests that long, near-deterministic stretches are unlikely to be efficiency-optimal, and directly motivates a segment-aware trimming principle. We then instantiate this principle in RL via trimming-based reward shaping that penalizes the identified high-probability segments, steering the policy toward more concise yet faithful CoTs.

### 3.1. Problem Formulation and Theoretical Insight

Following the notation in Section 2, let $L(z)$ denote the number of tokens in $z$. Our goal is to minimize the expected length of the reasoning chain while ensuring that the correctness probability remains above a desired threshold. Formally, this can be expressed as the following constrained optimization problem:

$$\min_\theta \mathbb{E}_{z \sim \pi_\theta(\cdot|x)} \left[ L(z) \right],$$

$$\text{s.t.} \quad P_\theta(\mathcal{A} \mid x) = \mathbb{E}_{z \sim \pi_\theta(\cdot|x)} \left[ P_\theta(\mathcal{A} \mid x, z) \right] \geq p_0,$$

where $p_0 \in (0, 1)$ denotes the minimum acceptable accuracy level. This formulation captures the inherent trade-off between reasoning efficiency and solution accuracy in CoT generation. To facilitate optimization and analysis, we consider maximizing the corresponding Lagrangian form:

$$\mathcal{H}(\theta; \lambda) = \mathbb{E}_{z \sim \pi_\theta(\cdot|x)} \left[ P_\theta(\mathcal{A} \mid x, z) \right] - \lambda \, \mathbb{E}_{z \sim \pi_\theta(\cdot|x)} \left[ L(z) \right], \tag{6}$$

where $\lambda > 0$ serves as a length penalty coefficient that explicitly balances correctness and efficiency. This formulation provides a principled foundation for designing algorithms that encourage concise yet accurate reasoning in LLMs. A common instantiation in related work is to

incorporate reasoning length directly into the objective via a length-weighted term, using $\lambda$ to regulate the pressure toward shorter CoTs (Hou et al., 2025; Aggarwal & Welleck, 2025; Liu et al., 2025a). However, such token-uniform regularization does not account for the internal structure of a reasoning trace. We therefore take a segment-level view and ask how allocating probability mass to generate a particular segment affects $\mathcal{H}(\theta; \lambda)$, compared to alternative continuations. The following proposition then provides a segment-level sufficient condition under which generating a particular segment becomes suboptimal.

**Proposition 3.1** (A sufficient condition for segment suboptimality). *Let $x$ be an input and let $z \sim \pi_\theta(\cdot \mid x)$ denote a complete reasoning trace. Consider a candidate segment $z_1$ that occurs within $z$. Without loss of generality, we assume $z_1$ is a prefix of the trace (any preceding context can be absorbed into $x$), and write*

$$z = [\, z_1 \| z_2 \,], \qquad (7)$$

*where $\|$ denotes token concatenation and $z_2$ is the continuation sampled from $\pi_\theta(\cdot \mid x, z_1)$. Assume $0 < \pi_\theta(z_1 \mid x) < 1$. If*

$$L(z_1) + \mathbb{E}_{z_2 \sim \pi_\theta(\cdot \mid x, z_1)}[L(z_2)] - \mathbb{E}_{z' \sim \pi_\theta(\cdot \mid x)}[L(z')]$$
$$> \frac{1}{\lambda}\left(\frac{1}{\pi_\theta(z_1 \mid x)} - 1\right) P_\theta(\mathcal{A} \mid x), \qquad (8)$$

*where $z'$ is a dummy variable denoting a generic trace sampled from $\pi_\theta(\cdot \mid x)$, then assigning probability mass to generating $z_1$ is suboptimal under the correctness-length objective: decreasing $\pi_\theta(z_1 \mid x)$ (and redistributing the removed mass to alternative traces) strictly increases the objective.*

The proof and further discussion are provided in Appendix A.1 and B.1. Proposition 3.1 implies a sufficient condition under which allocating probability mass to a segment $z_1$ is suboptimal under the correctness-length objective. In particular, the condition becomes easier to satisfy when $z_1$ is length-increasing and generated with high probability: longer $z_1$ directly enlarges the left-hand side through the additive term $L(z_1)$, while higher $\pi_\theta(z_1 \mid x)$ shrinks the right-hand side. Consequently, long yet high-confidence stretches are natural candidates for suppression. This observation motivates a simple yet principled inductive bias for efficient reasoning: suppress segments that are both length-increasing and near-deterministic.

### 3.2. Segment-aware Trimming via Reward Shaping

Guided by the theoretical insight in Proposition 3.1, our goal is to discourage *redundant* portions of the reasoning trace that are locally high-probability under the current policy and persist for a nontrivial span. We implement this idea with a simple segment-level reward shaping rule that detects and penalizes *long high-probability stretches* using a sliding-window proxy.

**Sliding-window high-probability count.** For each rollout, we consider the CoT token sequence $z = (z_1, \ldots, z_T)$ sampled from the policy $\pi_{\text{old}}$. We traverse $z$ with a fixed sliding window of size $w$. For each window starting at position $t$, its joint probability under $\pi_{\text{old}}$ is

$$\pi_{\text{old}}(z_{t:t+w-1} \mid x, z_{<t}) = \prod_{j=t}^{t+w-1} \pi_{\text{old}}(z_j \mid x, z_{<j}), \quad (9)$$

where $t = 1, \ldots, T - w + 1$. We count the number of windows whose joint probability exceeds a threshold $\tau$:

$$C(z) = \sum_{t=1}^{T-w+1} \mathbb{I}[\pi_{\text{old}}(z_{t:t+w-1} \mid x, z_{<t}) \geq \tau], \quad (10)$$

and use this count to define the efficient-reasoning reward.

Crucially, Eq. (10) is designed to operationalize Proposition 3.1: the overlapping window count $C(z)$ is a direct, computable proxy for identifying stretches that are simultaneously *high-probability* and *long*. If a redundant high-probability stretch of length $L$ exists ($L > w$), then it induces $L - w + 1$ *overlapping* windows whose probabilities all exceed the threshold. Hence $C(z)$ grows roughly *linearly* with the length of such a stretch, serving as a practical surrogate for the theoretical condition.

**Correctness-gated trimming reward.** The trimming reward is applied only to positive rollouts (i.e., those judged correct by the rule-based verifier), following a common design choice in prior efficiency-oriented RL. Formally, we define

$$r_{\text{ER}}(z) = \begin{cases} -C(z), & \text{if the rollout is correct,} \\ 0, & \text{otherwise.} \end{cases} \quad (11)$$

This choice prevents the efficiency term from encouraging degenerate shortcuts on incorrect trajectories and focuses the trimming pressure on reducing computational inefficiency among correct solutions.

**Reward composition.** We combine the original reward and the trimming reward as

$$r = r_{\text{ori}} + \lambda \, r_{\text{ER}}, \qquad (12)$$

where $\lambda$ controls the strength of the trimming signal.

**Practical notes.** Although SLAT involves $(w, \tau, \lambda)$, in practice it behaves like a **one knob method**. These hyperparameters all serve the same purpose, controlling how

strongly we trim and compress the reasoning trace. In practice, we fix $\tau = 0.9$ throughout and use $w$ to control how aggressive the detector is. A larger $w$ is more selective and flags only long high probability stretches. We set $\lambda = 0.001$, so that $\lambda r_{ER}$ is typically about one order of magnitude smaller than the scale of $r_{ori}$, to keep the trimming term a gentle regularizer. We implement this segment-level adaptive trimming scheme as SLAT (**S**egment **L**evel **A**daptive **T**rimming).

## 4. Experiment

We conduct experiments primarily on mathematical reasoning benchmarks, reporting both correctness and reasoning length under a unified decoding protocol. We consider two main settings: compressing CoTs on a strong distilled reasoner (`DeepSeek-R1-Distill-Qwen-7B`) and training from a base model (`Qwen2.5-Math-7B`) for reasoning while controlling CoT length. We also report extensions to additional backbones (`Qwen3-14B`) and additional analyses on out-of-domain generalization and training efficiency.

### 4.1. Setting

**Models.** For the distilled-reasoner setting, we apply SLAT to `DeepSeek-R1-Distill-Qwen-7B` (Guo et al., 2025) and benchmark it against state-of-the-art released models trained on the same base model by directly evaluating their publicly available checkpoints under the same protocol. For the base-model training setting, we train `Qwen2.5-Math-7B` (Yang et al., 2024) with GRPO and compare SLAT to representative length-reward designs, including `Vanilla Truncation` and the objectives proposed in `Kimi-k1.5` and `TMLRE`, all implemented under the same GRPO recipe. To assess generalization to larger models, we additionally experiment with `Qwen3-14B` (Yang et al., 2025a).

**Evaluation** We focus on mathematical reasoning and evaluate on five benchmarks: MATH500 (Hendrycks et al., 2021), OlympiadBench (He et al., 2024), AMC23 (MAA, 2023), AIME24&25 (MAA). For all models, we sample with temperature 0.7 and top-p=1. We report accuracy as avg@4 (avg@k means the average correctness score over k samples) on MATH500 and Olympiad Bench, and as avg@16 on AMC23, AIME24, and AIME25 due to their limited number of test problems. In addition, we report CoT length statistics to measure reasoning efficiency. To assess out-of-domain generalization, we additionally evaluate on MMLU-STEM (Hendrycks et al., 2020) and GPQA-Diamond (Rein et al., 2024) datasets. Other evaluation details are provided in Appendix C.

**Training Setup** All training runs are conducted on NVIDIA A100 GPUs using the **verl** (Sheng et al., 2025) framework, with DAPO-MATH-17K (Yu et al., 2025) as the training dataset. For SLAT, we follow the GRPO recipe in Section 2 with minor modifications and incorporate the efficient-reasoning reward defined in Section 3.2. Full hyperparameters and implementation details are deferred to Appendix C.

### 4.2. Main Results

#### 4.2.1. COMPRESSING CoTs ON DISTILLED MODELS

We study the setting of compressing CoTs on a strong distilled reasoner, where the model already exhibits strong reasoning behavior and the goal is to improve efficiency without sacrificing correctness. We compare SLAT against several representative released baselines, including `LC-R1` (Cheng et al., 2025), `TLMRE` with different compression coefficients $\alpha$ (Arora & Zanette, 2025), `AdaptThink` (Zhang et al., 2025b), `DAST` (Shen et al., 2025) and `L1-Max` (Aggarwal & Welleck, 2025). Table 1 reports the accuracy-length results, where SLAT consistently improves both correctness and efficiency, outperforming 6 of 7 baselines with higher accuracy and shorter CoTs, i.e., a Pareto improvement on average in the accuracy-length trade-off. In particular, SLAT attains the best average accuracy (66.4) while reducing the average CoT length from 8404 to 4176 tokens (a 50% reduction) compared to the original model. Relative to the baselines `LC-R1`, `TLMRE`, and `AdaptThink`, SLAT improves average accuracy by up to 2.4 points while reducing CoT length. `DAST` is competitive in accuracy but offers only modest length savings. Furthermore, compared to the aggressive compression baseline `L1-Max`, SLAT demonstrates a superior accuracy-efficiency trade-off, achieving significantly higher accuracy (+5.2 points) with only a moderate increase in sequence length.

#### 4.2.2. COMPRESSING CoTs IN BASE-MODEL TRAINING

We next study a controlled setting on `Qwen2.5-Math-7B`, where reasoning ability is learned via RL and efficiency is also shaped during training. As length-oriented baselines, we implement representative length-objective (reward) designs from prior work, including vanilla `Truncation` and the objective provided in `Kimi-k1.5` (Team et al., 2025b) (denoted as `Length-MaxMin`) and `TLMRE` (denoted as `Length-Mean`), and apply them within the same GRPO training recipe for a fair comparison. For SLAT, we vary the window size $w$ and the weighting coefficient $\lambda$ to control the trimming strength. Figure 2 visualizes the accuracy-length trade-off on `Qwen2.5-Math-7B` across math benchmarks, with detailed numerical results provided in Appendix D.2. While GRPO maximizes accuracy, it

*Table 1.* Performance comparison of methods based on `DeepSeek-R1-Distill-Qwen-7B` (denoted as Original). We report accuracy (Acc.) and average token length (Len.) across datasets. Baselines utilize official checkpoints evaluated under a unified protocol. The final row quantifies the relative change (%) of SLAT compared to the original model.

| | MATH 500 | | Olympiad Bench | | AMC23 | | AIME24 | | AIME25 | | Avg. | |
|---|---|---|---|---|---|---|---|---|---|---|---|---|
| | Acc. | Len. | Acc. | Len. | Acc. | Len. | Acc. | Len. | Acc. | Len. | Acc.↑ | Len.↓ |
| Original | 91.9 | 3860 | 59.1 | 8093 | 87.8 | 6001 | 53.8 | 11731 | **37.6** | 12334 | 66.0 | 8404 |
| LC-R1 | 89.8 | **1583** | 57.3 | 4156 | 84.8 | 3045 | 52.3 | 6835 | 35.9 | 7761 | 64.0 | 4676 |
| AdaptThink | 91.6 | 1893 | 59.1 | 5945 | 85.3 | 3740 | **54.2** | 9565 | 37.4 | 10217 | 65.5 | 6272 |
| DAST | 92.4 | 3228 | 58.3 | 7390 | 88.6 | 5224 | 54.3 | 10924 | 36.9 | 11552 | 66.1 | 7664 |
| TLMRE$_{\alpha=0.1}$ | 91.2 | 2662 | 58.0 | 6289 | 87.8 | 4405 | 52.8 | 9454 | 36.5 | 10650 | 65.3 | 6692 |
| TLMRE$_{\alpha=0.2}$ | 90.7 | 2312 | 57.2 | 5679 | 88.3 | 4135 | 51.2 | 8987 | 36.9 | 9478 | 64.9 | 6118 |
| L1-Max | 90.4 | 2134 | 55.8 | **2854** | 85.0 | **2626** | 42.9 | **4135** | 31.7 | **3970** | 61.2 | **3144** |
| SLAT | **92.8** | 1820 | **59.6** | 3714 | **89.1** | 2857 | 52.9 | 6105 | **37.6** | 6385 | **66.4** | 4176 |
| | +1% | -53% | +1% | -54% | +1.5% | -52% | -1.7% | -48% | +0% | -48% | +0.6% | -50% |

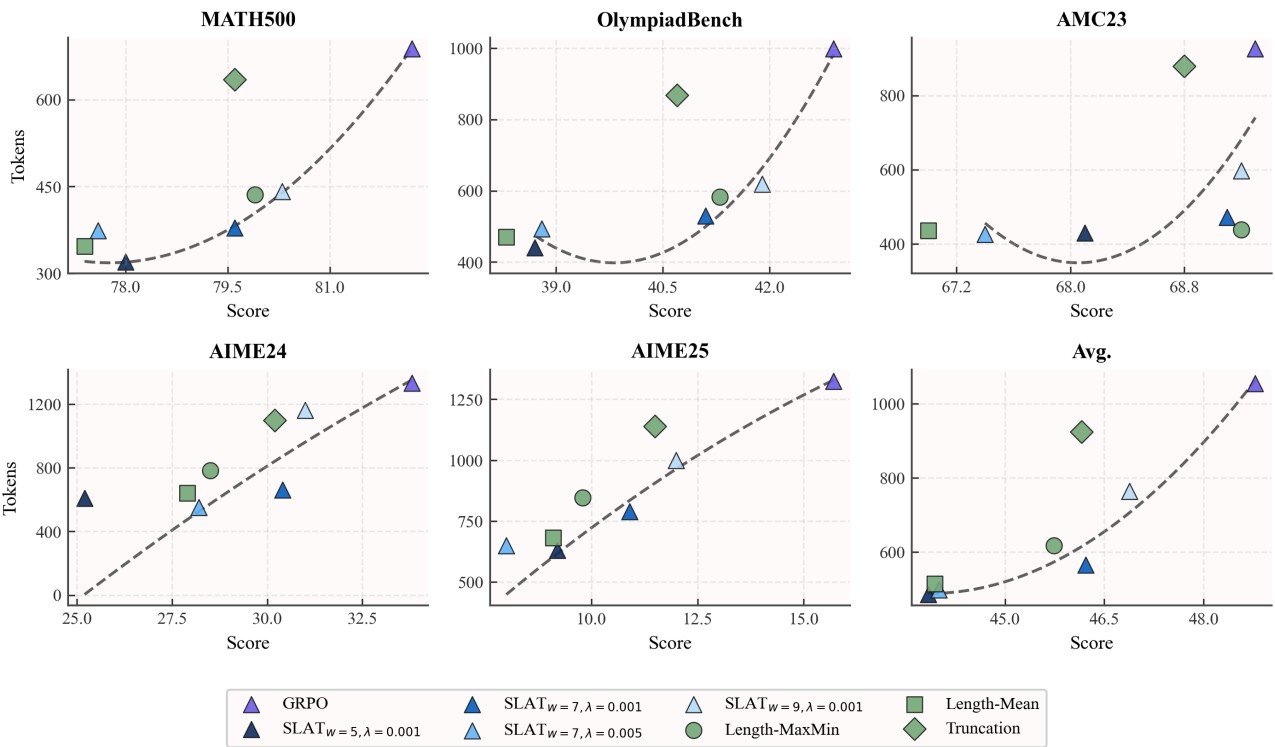

*Figure 2.* Accuracy-length trade-off on math benchmarks for models trained from the same base model `Qwen2.5-Math-7B` under different training objectives. Each point corresponds to a specific training objective, including SLAT variants obtained by varying the window size $w$ and coefficient $\lambda$. Across benchmarks, SLAT typically provides a more favorable accuracy-length profile than token-uniform length objectives, especially on harder datasets.

results in inefficient, lengthy chains. SLAT establishes a superior trade-off curve, allowing precise modulation of inference cost versus performance via $w$ and $\lambda$. Among these baselines, `Truncation` performs relatively poorly across benchmarks, suggesting that naive truncation alone is insufficient for balancing accuracy and length during rea-

soning training. In particular, length-based objectives can work well on simpler or medium-difficulty datasets, for example `Length-MaxMin` performs strongly on AMC23. However, on higher-difficulty benchmarks such as AIME24 and AIME25, SLAT tends to offer clearer advantages, suggesting that global penalties disrupt long-horizon reasoning

dependencies, whereas SLAT's local, segment-aware trimming preserves the critical logical structures required for complex problem solving.

### 4.3. Additional Analyses

#### 4.3.1. EXTENSION TO LARGER MODEL

Table 2 reports results on `Qwen3-14B`. As shown in the table, applying GRPO substantially improves mathematical accuracy on this larger base model but also leads to much longer CoTs (e.g., average length 5024 tokens). When adding SLAT on top of GRPO, we obtain a markedly more efficient model: the average CoT length is reduced from 5024 to 2677 tokens ($-47\%$) while keeping accuracy nearly unchanged on average ($-0.5\%$). Notably, on simpler to medium-difficulty benchmarks, SLAT can compress CoTs without hurting correctness and even improves performance on AMC23 ($+3.4\%$ with a $-61\%$ length reduction), indicating that SLAT scales well to larger base models.

#### 4.3.2. GENERALIZATION BEYOND MATH

While our primary focus is mathematical reasoning, we additionally evaluate cross-domain generalization on MMLU-STEM and GPQA-Diamond for models trained on `DeepSeek-R1-Distill-Qwen-7B`. As shown in Table 6 in Appendix, on MMLU-STEM, SLAT matches the strongest baseline in accuracy while achieving the shortest CoT length, and on GPQA-Diamond, it attains the competitive accuracy among the compared methods with a clear length reduction compared to the original checkpoint. These results suggest that SLAT generalizes reasonably well beyond mathematical reasoning.

#### 4.3.3. TRAINING EFFICIENCY

We further analyze the impact of SLAT on training time. Beyond its modest per-step overhead, SLAT can reduce overall training time by encouraging shorter rollouts in later training stages, thereby reducing generation cost and improving training throughput. Detailed throughput and wall-clock comparisons are provided in Figure 3. As shown in the figure, GRPO exhibits relatively stable per-step time and maintains longer generations as training proceeds. In contrast, SLAT drives a sustained reduction in response length in later stages, which in turn lowers the per-step wall-clock time and yields faster training overall.

#### 4.3.4. QUALITATIVE ANALYSIS OF $2+3$

To complement our quantitative results, we provide a qualitative comparison between the original distilled reasoner (`DeepSeek-R1-Distill-Qwen-7B`) and its SLAT-trained counterpart. As illustrated in Appendix D.4 (Fig. 6). The baseline model often exhibits *overthinking* even on

trivial queries: after reaching the correct computation, it continues with verbose scaffolding (restating the problem, invoking generic principles and enumerating step-by-step instructions). In contrast, SLAT produces a substantially shorter trace that preserves the essential reasoning and terminates promptly once the answer is determined. This qualitative behavioral analysis corroborates our primary hypothesis that SLAT enhances inference efficiency while maintaining logical fidelity. We also provide qualitative analysis on question from MATH500 in Appendix D.4.

## 5. Related Work

**Overthinking and Halting in Chain-of-Thought Reasoning** CoT prompting markedly improves reasoning by externalizing intermediate steps (Wei et al., 2022), but it can also induce redundancy and overthinking, where models keep elaborating after a valid solution emerges (Chen et al., 2024). Consequently, a stream of when-to-stop methods seeks to cut reasoning at the point of diminishing returns. Laaouach (2025) stops when token-level uncertainty stabilizes, saving tokens on arithmetic tasks. Wei et al. (2025) halt at the first coherent solution boundary to avoid trailing repetition. Beyond halting, compression-based approaches shorten the trace itself: TokenSkip skips low-importance tokens for controllable CoT compression (Xia et al., 2025), while C3oT produces condensed rationales via compression-conditioned prompting (Kang et al., 2025). Recent analyses further show that overly long CoTs can propagate errors and hurt performance on easier instances (Yeo et al., 2025), and surveys summarize the broader efficiency-accuracy tradeoff (Sui et al., 2025; Feng et al., 2025). Collectively, these findings frame efficient reasoning as knowing when to stop, not merely thinking longer.

**Reinforcement Learning for Chain-of-Thought Optimization** RL is increasingly used to shape the reasoning process itself, with efficiency encoded directly in the objective rather than added post hoc. We group prior work into three lines.

*Length aware objectives* make brevity explicit in training. ThinkPrune(Hou et al., 2025) casts reasoning as a budgeted objective, where over-budget traces receive zero credit, and a curriculum of shrinking caps teaches concise completion. Conversely, O1-Pruner (Luo et al., 2025) and LC-R1 (Cheng et al., 2025) incorporate length as a soft regularization term, optimizing the Pareto frontier between accuracy and token efficiency. Similarly, controllable frameworks like L1 (Aggarwal & Welleck, 2025) and LASER (Liu et al., 2025a) parameterize sequence length as a conditional target to enforce strict budget adherence.

*Information or uncertainty-driven trimming and halting* utilizes entropy dynamics to govern inference capability.

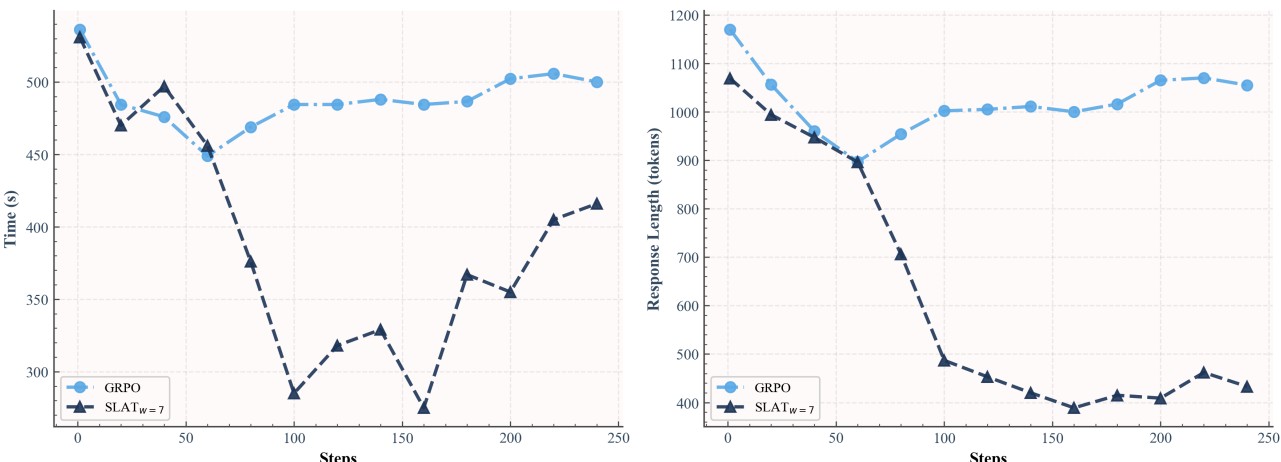

*Figure 3.* Training efficiency comparison between vanilla GRPO and SLAT. We plot the per-step wall-clock time (top) and the average response length (bottom) over training steps, showing that SLAT encourages shorter rollouts in later stages and consequently reduces training time.

*Table 2.* Extension of SLAT to the training of `Qwen3-14B`. Accuracy (Acc.) and average token length (Len.) are reported for each dataset. The last row additionally reports the relative change (%) of SLAT with respect to GRPO.

| | MATH 500 | | Olympiad Bench | | AMC23 | | AIME24 | | AIME25 | | Avg. | |
|---|---|---|---|---|---|---|---|---|---|---|---|---|
| | Acc. | Len. | Acc. | Len. | Acc. | Len. | Acc. | Len. | Acc. | Len. | Acc.↑ | Len.↓ |
| Qwen3-14B | 65.1 | 655 | 35.8 | 1084 | 56.7 | 863 | 28.8 | 1641 | 7.5 | 1835 | 38.8 | 1216 |
| + GRPO | 87.1 | 2978 | 55.2 | 4824 | 82.8 | 4317 | 46.5 | 6362 | 33.1 | 6638 | 60.9 | 5024 |
| + SLAT | 86.2 | 901 | 54.8 | 2106 | 85.6 | 1704 | 45.0 | 4198 | 31.6 | 4477 | 60.6 | 2677 |
| | -1.0% | -70% | -0.7% | -56% | +3.4% | -61% | -3.2% | -34% | -4.5% | -33% | -0.5% | -47% |

From an uncertainty-based perspective, step-level entropy identifies low-information segments that can be skipped (Li et al., 2025), and cumulative entropy suggests stopping once uncertainty plateaus (Jiang et al., 2025).

*Adaptive allocation of compute* rests on the observation that different instances require different amounts of reasoning, with supporting analyses such as token complexity, which formalizes a minimal token budget per instance (Lee et al., 2025). Building on these signals, adaptive methods adjust computation to the instance or to external intent. Instance adaptive methods modulate pressure by difficulty so that easy inputs receive less computation while hard inputs retain deeper reasoning (Ling et al., 2025; Shen et al., 2025; Zhang et al., 2025b; Arora & Zanette, 2025). Preference-based variants encourage correct yet shorter traces through comparative feedback (Yang et al., 2025b). Controllable formulations align user intent with compute, for example by learning discrete reasoning regimes, as in ThinkDial (He et al., 2025), or by selecting among output formats, as in ARM (Wu et al., 2025). While such user-steerable interfaces are practical for deployment, they rely on external interven-

tion and thus fall short of *autonomous* efficiency, ultimately contradicting the goal of achieving fully autonomous AI.

Situated within this landscape, we adopt a structure-aware view of efficiency. Rather than penalizing length uniformly, we analyze the CoT at the *segment level* and establish, under an explicit objective that balances correctness and length, a sufficient condition that identifies high-probability, low-information segments as suboptimal. This theoretical lens motivates a trimming bias within reinforcement learning, where the policy is encouraged to suppress segments that satisfy the condition while preserving decision-relevant steps. The formulation offers a localized intervention guided by an explicit condition, complementing sequence-level length control and uncertainty-based heuristics without prescribing a fixed budget or mode.

## 6. Conclusion and Future Work

We studied efficient Chain-of-Thought reasoning through the lens of segment-level redundancy. Motivated by qualitative observations of over-elaboration in long traces, we

formulated an explicit correctness-length objective and derived a sufficient condition suggesting that certain long, near-deterministic segments are unlikely to be favored under an efficiency-optimal policy. Based on this insight, we proposed SLAT, a simple yet effective reward-shaping approach that penalizes high-probability stretches during RL training to suppress redundant computation while maintaining correctness. Across strong distilled reasoners and base models trained with GRPO, SLAT yields a consistently better accuracy-length profile on mathematical reasoning benchmarks. Our extensions to larger base models and additional domains further suggest that the approach remains effective as model capacity and task scope increase.

There are several promising directions for future work. First, while SLAT is already effective with a fixed sliding-window criterion, incorporating more adaptive mechanisms to identify redundant stretches could further improve robustness and reduce reliance on manual hyperparameter choices. Second, an important next step is to scale SLAT to substantially larger model families (100B+ parameters) and to evaluate whether the same accuracy-length gains persist at scale. It would also be valuable to examine whether SLAT generalizes to broader reasoning scenarios, such as domain-specialized DeepResearch agents (Wang et al., 2026b). Third, beyond efficiency and accuracy, the safety implications of reasoning compression deserve further investigation. Although SLAT targets redundant reasoning segments rather than model parameters or visual tokens, understanding whether training-time trimming affects adversarial robustness, information leakage, or model extraction risks is important for safe deployment (Zhang et al., 2025d; 2026). Finally, combining training-time trimming with complementary inference-time efficiency techniques may yield additional speedups without sacrificing correctness.

## Acknowledgments

This work was supported in part by the Research Grants Council of the Hong Kong SAR (Grant No. C5052-23G, PolyU15229824, SRFS2526- 5S04), and The Hong Kong Polytechnic University (Project IDs: P0051130, P0060651, P0058445). This work was supported in part by Guangdong Basic and Applied Basic Research Foundation (No. 2024B1515020019).

## Impact Statement

This paper presents work whose goal is to advance the field of LLM. There are many potential societal consequences of our work, none of which we feel must be specifically highlighted here.

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

# A. Theoretical Analysis

## A.1. Proof of Proposition 3.1

**Proposition A.1** (Restatement of Proposition 3.1). *Let $x$ be an input and let $z \sim \pi_\theta(\cdot \mid x)$ denote a complete reasoning trace. Consider a candidate segment $z_1$ that occurs within $z$. Without loss of generality, we assume $z_1$ is a prefix of the trace (any preceding context can be absorbed into $x$), and write*

$$z = [\, z_1 \| z_2 \,], \tag{13}$$

*where $\|$ denotes token concatenation and $z_2$ is the continuation sampled from $\pi_\theta(\cdot \mid x, z_1)$. Assume $0 < \pi_\theta(z_1 \mid x) < 1$. If*

$$L(z_1) + \mathbb{E}_{z_2 \sim \pi_\theta(\cdot \mid x, z_1)}[L(z_2)] - \mathbb{E}_{z' \sim \pi_\theta(\cdot \mid x)}[L(z')] > \frac{1}{\lambda}\left(\frac{1}{\pi_\theta(z_1 \mid x)} - 1\right) P_\theta(\mathcal{A} \mid x), \tag{14}$$

*where $z'$ is a dummy variable denoting a generic trace sampled from $\pi_\theta(\cdot \mid x)$, then assigning probability mass to generating $z_1$ is suboptimal under the correctness-length objective: decreasing $\pi_\theta(z_1 \mid x)$ (and redistributing the removed mass to alternative traces) strictly increases the objective.*

*Proof.* We begin by rewriting the objective in Eq. 6 by partitioning on whether the prefix $z_1$ is generated:

$$\mathcal{H}(\theta; \lambda) = \pi_\theta(z_1 \mid x)\Big\{\mathbb{E}_{z_2 \sim \pi_\theta(\cdot \mid x, z_1)}\big[P_\theta(\mathcal{A} \mid x, z_1, z_2)\big] - \lambda L(z_1) - \lambda \mathbb{E}_{z_2 \sim \pi_\theta(\cdot \mid x, z_1)}[L(z_2)]\Big\}$$
$$+ \big(1 - \pi_\theta(z_1 \mid x)\big)\Big\{\mathbb{E}_{z_1', z_2' \sim \pi_\theta(\cdot \mid x)\,;\, z_1' \neq z_1}\big[P_\theta(\mathcal{A} \mid x, z_1', z_2')\big] - \lambda \mathbb{E}_{z_1', z_2' \sim \pi_\theta(\cdot \mid x)\,;\, z_1' \neq z_1}\big[L(z')\big]\Big\}, \tag{15}$$

where $z' = [z_1' \| z_2']$ denotes an alternative reasoning trace. This partition is introduced solely to isolate the contribution of the branch that begins with $z_1$.

Our goal is to characterize when the branch that begins with $z_1$ is strictly worse than the alternative branch:

$$\mathbb{E}_{z_2 \sim \pi_\theta(\cdot \mid x, z_1)}\big[P_\theta(\mathcal{A} \mid x, z_1, z_2)\big] - \lambda L(z_1) - \lambda \mathbb{E}_{z_2 \sim \pi_\theta(\cdot \mid x, z_1)}\big[L(z_2)\big]$$
$$< \mathbb{E}_{z_1', z_2' \sim \pi_\theta(\cdot \mid x)\,;\, z_1' \neq z_1}\big[P_\theta(\mathcal{A} \mid x, z_1', z_2')\big] - \lambda \mathbb{E}_{z_1', z_2' \sim \pi_\theta(\cdot \mid x)\,;\, z_1' \neq z_1}\big[L(z')\big]. \tag{16}$$

If Eq. (16) holds, then decreasing the probability mass assigned to generating $z_1$ (i.e., shifting probability away from the branch that starts with $z_1$) strictly increases $\mathcal{H}(\theta; \lambda)$.

**Step 1:** To evaluate the right-hand side of Eq. (16), we first express expectations conditioned on $z_1' \neq z_1$ in terms of unconditional expectations. For any measurable function $f(x, z_1', z_2')$, we have

$$\mathbb{E}_{z_1', z_2' \sim \pi_\theta(\cdot \mid x)\,;\, z_1' \neq z_1}\big[f(x, z_1', z_2')\big]$$
$$= \sum_{z_1' \neq z_1} \sum_{z_2'} f(x, z_1', z_2')\, \pi_\theta(z_2' \mid x, z_1')\, \frac{\pi_\theta(z_1' \mid x)}{1 - \pi_\theta(z_1 \mid x)}$$
$$= \frac{1}{1 - \pi_\theta(z_1 \mid x)}\Bigg[\sum_{z_1', z_2'} f(x, z_1', z_2')\, \pi_\theta(z_2' \mid x, z_1')\, \pi_\theta(z_1' \mid x) - \pi_\theta(z_1 \mid x) \sum_{z_2'} f(x, z_1, z_2')\, \pi_\theta(z_2' \mid x, z_1)\Bigg]$$
$$= \frac{1}{1 - \pi_\theta(z_1 \mid x)}\Big[\mathbb{E}_{z_1', z_2' \sim \pi_\theta(\cdot \mid x)}\big[f(x, z_1', z_2')\big] - \pi_\theta(z_1 \mid x)\, \mathbb{E}_{z_2 \sim \pi_\theta(\cdot \mid x, z_1)}\big[f(x, z_1, z_2)\big]\Big]$$
$$= \frac{1}{1 - \pi_\theta(z_1 \mid x)}\Big[\mathbb{E}_{z' \sim \pi_\theta(\cdot \mid x)}\big[f(x, z')\big] - \pi_\theta(z_1 \mid x)\, \mathbb{E}_{z_2 \sim \pi_\theta(\cdot \mid x, z_1)}\big[f(x, z_1, z_2)\big]\Big], \tag{17}$$

where $z' = [z_1' \| z_2']$. The second equality uses the definition of the conditional distribution given $z_1' \neq z_1$; the third equality follows by adding and subtracting the $(z_1' = z_1)$ term and regrouping.

Applying Eq. (17) to the right-hand side of Eq. (16) yields

$$
\begin{aligned}
&\mathbb{E}_{z_1', z_2' \sim \pi_\theta(\cdot|x)\,;\, z_1' \neq z_1}\big[P_\theta(\mathcal{A} \mid x, z_1', z_2')\big] - \lambda \, \mathbb{E}_{z_1', z_2' \sim \pi_\theta(\cdot|x)\,;\, z_1' \neq z_1}\big[L(z')\big] \\
&= \frac{1}{1 - \pi_\theta(z_1 \mid x)} \Big\{ \mathbb{E}_{z' \sim \pi_\theta(\cdot|x)}\big[P_\theta(\mathcal{A} \mid x, z')\big] - \lambda \, \mathbb{E}_{z' \sim \pi_\theta(\cdot|x)}\big[L(z')\big] \\
&\qquad - \pi_\theta(z_1 \mid x) \, \mathbb{E}_{z_2 \sim \pi_\theta(\cdot|x,z_1)}\big[P_\theta(\mathcal{A} \mid x, z_1, z_2) - \lambda\big(L(z_1) + L(z_2)\big)\big] \Big\}.
\end{aligned}
\tag{18}
$$

**Step 2:** Substituting Eq. (18) into the right-hand side of Eq. (16) gives

$$
\begin{aligned}
&\mathbb{E}_{z_2 \sim \pi_\theta(\cdot|x,z_1)}\big[P_\theta(\mathcal{A} \mid x, z_1, z_2)\big] - \lambda L(z_1) - \lambda \mathbb{E}_{z_2 \sim \pi_\theta(\cdot|x,z_1)}\big[L(z_2)\big] \\
&< \frac{1}{1 - \pi_\theta(z_1 \mid x)} \Big\{ \mathbb{E}_{z' \sim \pi_\theta(\cdot|x)}\big[P_\theta(\mathcal{A} \mid x, z')\big] - \lambda \mathbb{E}_{z' \sim \pi_\theta(\cdot|x)}\big[L(z')\big] \\
&\qquad - \pi_\theta(z_1 \mid x) \, \mathbb{E}_{z_2 \sim \pi_\theta(\cdot|x,z_1)}\big[P_\theta(\mathcal{A} \mid x, z_1, z_2) - \lambda\big(L(z_1) + L(z_2)\big)\big] \Big\}.
\end{aligned}
\tag{19}
$$

Multiplying both sides by $1 - \pi_\theta(z_1 \mid x)$ and expanding the right-hand side yield

$$
\begin{aligned}
&\big(1 - \pi_\theta(z_1 \mid x)\big) \mathbb{E}_{z_2 \sim \pi_\theta(\cdot|x,z_1)}\big[P_\theta(\mathcal{A} \mid x, z_1, z_2)\big] - \big(1 - \pi_\theta(z_1 \mid x)\big)\lambda L(z_1) \\
&\quad - \big(1 - \pi_\theta(z_1 \mid x)\big)\lambda \, \mathbb{E}_{z_2 \sim \pi_\theta(\cdot|x,z_1)}\big[L(z_2)\big] \\
&< \mathbb{E}_{z' \sim \pi_\theta(\cdot|x)}\big[P_\theta(\mathcal{A} \mid x, z')\big] - \lambda \, \mathbb{E}_{z' \sim \pi_\theta(\cdot|x)}\big[L(z')\big] - \pi_\theta(z_1 \mid x) \, \mathbb{E}_{z_2 \sim \pi_\theta(\cdot|x,z_1)}\big[P_\theta(\mathcal{A} \mid x, z_1, z_2)\big] \\
&\quad + \pi_\theta(z_1 \mid x)\lambda L(z_1) + \pi_\theta(z_1 \mid x)\lambda \, \mathbb{E}_{z_2 \sim \pi_\theta(\cdot|x,z_1)}\big[L(z_2)\big].
\end{aligned}
\tag{20}
$$

Now move all terms to the left-hand side and cancel the matching $\pi_\theta(z_1 \mid x)$ terms; this gives

$$
\begin{aligned}
&\mathbb{E}_{z_2 \sim \pi_\theta(\cdot|x,z_1)}[P_\theta(\mathcal{A} \mid x, z_1, z_2)] - \mathbb{E}_{z' \sim \pi_\theta(\cdot|x)}[P_\theta(\mathcal{A} \mid x, z')] - \lambda L(z_1) - \lambda \mathbb{E}_{z_2 \sim \pi_\theta(\cdot|x,z_1)}[L(z_2)] \\
&\quad + \lambda \mathbb{E}_{z' \sim \pi_\theta(\cdot|x)}[L(z')] < 0.
\end{aligned}
\tag{21}
$$

Applying the law of total expectation, i.e.,

$$
\mathbb{E}_{z_2 \sim \pi_\theta(\cdot|x,z_1)}\big[P_\theta(\mathcal{A} \mid x, z_1, z_2)\big] = P_\theta(\mathcal{A} \mid x, z_1), \qquad \mathbb{E}_{z' \sim \pi_\theta(\cdot|x)}\big[P_\theta(\mathcal{A} \mid x, z')\big] = P_\theta(\mathcal{A} \mid x),
$$

the inequality further simplifies to

$$
P_\theta(\mathcal{A} \mid x, z_1) - P_\theta(\mathcal{A} \mid x) - \lambda L(z_1) - \lambda \mathbb{E}_{z_2 \sim \pi_\theta(\cdot|x,z_1)}\big[L(z_2)\big] + \lambda \mathbb{E}_{z' \sim \pi_\theta(\cdot|x)}\big[L(z')\big] < 0.
\tag{22}
$$

**Step 3:** Note that

$$
P_\theta(\mathcal{A} \mid x, z_1) = \frac{P_\theta(\mathcal{A}, z_1 \mid x)}{\pi_\theta(z_1 \mid x)} \leq \frac{P_\theta(\mathcal{A} \mid x)}{\pi_\theta(z_1 \mid x)},
\tag{23}
$$

hence,

$$
\begin{aligned}
&P_\theta(\mathcal{A} \mid x, z_1) - P_\theta(\mathcal{A} \mid x) - \lambda L(z_1) - \lambda \mathbb{E}_{z_2 \sim \pi_\theta(z_2|x,z_1)}[L(z_2)] + \lambda \mathbb{E}_{z' \sim \pi_\theta(z'|x)}[L(z')] \\
&\leq \frac{1 - \pi_\theta(z_1|x)}{\pi_\theta(z_1|x)} P_\theta(\mathcal{A} \mid x) - \lambda L(z_1) - \lambda \mathbb{E}_{z_2 \sim \pi_\theta(z_2|x,z_1)}[L(z_2)] + \lambda \mathbb{E}_{z' \sim \pi_\theta(z'|x)}[L(z')].
\end{aligned}
\tag{24}
$$

Therefore, it suffices to enforce a stronger condition that guarantees Eq. (22). Specifically, whenever

$$
L(z_1) + \mathbb{E}_{z_2 \sim \pi_\theta(\cdot|x,z_1)}\big[L(z_2)\big] - \mathbb{E}_{z' \sim \pi_\theta(\cdot|x)}\big[L(z')\big] > \frac{1}{\lambda}\Big(\frac{1}{\pi_\theta(z_1 \mid x)} - 1\Big) P_\theta(\mathcal{A} \mid x),
\tag{25}
$$

the right-hand side of Eq. (24) is non-positive, and hence Eq. (22) holds.

Since Steps 1 and 2 show that Eq. (22) implies Eq. (16), Eq. (25) is a sufficient condition for Eq. (16). Consequently, assigning probability mass to generating $z_1$ is suboptimal under $\mathcal{H}(\theta; \lambda)$, and the objective can be improved by decreasing $\pi_\theta(z_1 \mid x)$ (i.e., shifting probability away from the branch that starts with $z_1$). And we complete the proof. □

# B. Discussion

## B.1. A global viewpoint on Proposition 3.1.

Proposition 3.1 also admits a simple "global" specialization by taking the segment to be the entire trace. Specifically, let $z$ denote a complete reasoning trace sampled from $\pi_\theta(\cdot \mid x)$, and set $z_1 = z$ and $z_2 = \emptyset$ (so $L(z_2) = 0$). Then the length term on the left-hand side reduces to

$$L(z_1) + \mathbb{E}_{z_2 \sim \pi_\theta(\cdot|x,z_1)}[L(z_2)] - \mathbb{E}_{z' \sim \pi_\theta(\cdot|x)}[L(z')] = L(z) - \mathbb{E}_{z' \sim \pi_\theta(\cdot|x)}[L(z')].$$

As a result, a sufficient global condition for the trace $z$ to be suboptimal under the correctness-length objective is

$$L(z) - \mathbb{E}_{z' \sim \pi_\theta(\cdot|x)}[L(z')] \;>\; \frac{1}{\lambda}\Big(\frac{1}{\pi_\theta(z \mid x)} - 1\Big) P_\theta(\mathcal{A} \mid x), \tag{26}$$

in which case decreasing the probability mass assigned to generating the exact trace $z$ strictly improves the objective.

**Why a segment-level (local) characterization is still needed.** Although Eq. (26) provides a clean global specialization, it is not practically useful for guiding training or identifying redundancy. First, it depends on the probability of an *entire* trace, $\pi_\theta(z \mid x)$, which is typically exponentially small in length and highly instance-specific; as a result, the term $\left(\frac{1}{\pi_\theta(z|x)} - 1\right)$ becomes numerically extreme and difficult to estimate reliably. Second, the implied action (reducing the probability of generating the *entire* trace $z$) cannot provide a practical inductive bias for efficiency: it offers no guidance on *where* redundancy arises within the response and thus provides no actionable direction.

## B.2. Evaluation Protocol

We make every effort to ensure a fair evaluation. For distilled reasoning models, our baselines are drawn from published prior work with publicly available checkpoints. For training from base models, we re-implement uniform length-regularized objectives and keep all other training settings identical. To minimize confounding factors, all reported numbers come from our own local reproduction under a unified evaluation pipeline shared across methods and model variants. In particular, we standardize (i) the prompt format and answer extraction rules, (ii) the decoding configuration, (iii) the maximum generation budget, and (iv) the inference backend to ensure consistent tokenization and throughput.

## B.3. Broader Future Directions

Beyond the directions discussed in Section 6, SLAT also opens up several possibilities for integration with other post-training paradigms. One promising direction is to combine segment-level trimming with diversity-oriented reinforcement learning. While SLAT discourages long, high-confidence redundant stretches, diversity-promoting RL objectives (Yao et al., 2023; Wu et al., 2023; Wang et al., 2026a) could help maintain multiple valid reasoning trajectories and prevent the policy from collapsing into overly compressed but brittle reasoning patterns. This may lead to a better balance between efficiency, exploration, and robustness. Another interesting direction is to explore whether SLAT-induced efficiency can be incorporated into model merging pipelines (Zhou et al., 2026; Yang et al., 2026) , enabling efficient reasoning behaviors to be combined with other desirable capabilities acquired from different post-training recipes. **More broadly, these directions suggest that reasoning efficiency should not be treated as an isolated objective, but as a controllable dimension that can be jointly optimized with other demands.**

# C. Implementation Details

All training runs are conducted on NVIDIA A100 GPUs using the **verl** (Sheng et al., 2025) framework, with DAPO-MATH-17K (Yu et al., 2025) as the training dataset. For GRPO, we follow the standard recipe, except that we use a larger clipping ratio and eliminate the kl loss as suggested in DAPO (Yu et al., 2025). For SLAT, we adopt the same GRPO recipe and additionally incorporate the efficient-reasoning reward defined in Section 3.2. For length-objective baselines, we follow our GRPO recipe and replace the shaping term with the corresponding length-based reward defined in Appendix D.2. Detailed training hyperparameters are reported in Table 3.

For evaluation, we mainly focus on mathematical reasoning and use five benchmarks: MATH500 (Hendrycks et al., 2021), OlympiadBench (He et al., 2024), AMC23 (MAA, 2023), and AIME24/25 (MAA). For all models, we sample with

*Table 3.* Hyperparameter settings in training

| Hyperparameter | Value |
|---|---|
| *General settings* | |
| base models | `Qwen2.5-Math-7B`, `Qwen3-14B` `DeepSeek-R1-Distill-Qwen-7B` |
| dataset | DAPO-Math-17K |
| max prompt length | 2048 |
| max completion length | 20480 (8192 for 14B models) |
| training batchsize | 512 (256 for 14B models) |
| filter overlong prompts | True |
| num generations | 16 |
| use vllm | true |
| vllm gpu memory utilization | 0.6 |
| learning rate | 1e-6 |
| lr scheduler type | cosine |
| kl loss coef | 0.0 |
| clip ratio low | 0.2 |
| clip ratio high | 0.28 |
| training epochs | 5 |
| number of GPUs per node | 8 |
| number of nodes | 4 |
| *Length-objective* | |
| $\lambda$ | 0.25 |
| `SLAT` | |
| w | 7 |
| $\lambda$ | 0.001 |
| $\tau$ | 0.9 |

temperature 0.7 and top-$p$=1. To reduce variance, we report accuracy as avg@4 on MATH500 and OlympiadBench, and as avg@16 on AMC23, AIME24, and AIME25 due to their limited number of test problems. In addition, we report CoT length statistics to measure reasoning efficiency. We follow the evaluation codebase from Qwen2.5-Math[1]. Evaluation hyperparameters are reported in Table 4. To assess out-of-domain generalization, we additionally evaluate on MMLU-STEM (Hendrycks et al., 2020) and GPQA (Rein et al., 2024). We follow the same evaluation protocol and report both accuracy and reasoning length using avg@4.

*Table 4.* Hyperparameter settings in evaluation

| Hyperparameter | Value |
| --- | --- |
| *General settings* | |
| temperature | 0.7 |
| top p | 1.0 |
| number of sampling | 4 (16 for AMC23 and AIME24&25) |
| use vllm | true |
| vllm gpu memory utilization | 0.6 |

# D. Additional Results

## D.1. Additional Reasoning Trace Cases

We provide additional case studies to further illustrate how overthinking manifests as long, high-probability redundant segments in CoT reasoning.

**High-probability repetition of problem statements.** As shown in Fig. 4, when solving the grouping problem, the model devotes a substantial portion of its CoT to restating the prompt rather than advancing the derivation. For example, it reiterates the setup and constraints in near-verbatim form (e.g., "The problem requires us to group 3 men and 4 women into three groups with at least one man and one woman in each group," and "One group will be of size 3 and the other two will be of size 2 each"). The token-probability heatmap highlights these restatements as long contiguous high-probability blocks, indicating near-deterministic repetition with limited informational gain. This case illustrates an overthinking pattern where high-probability repetition of problem specifications inflates CoT length without contributing new reasoning content.

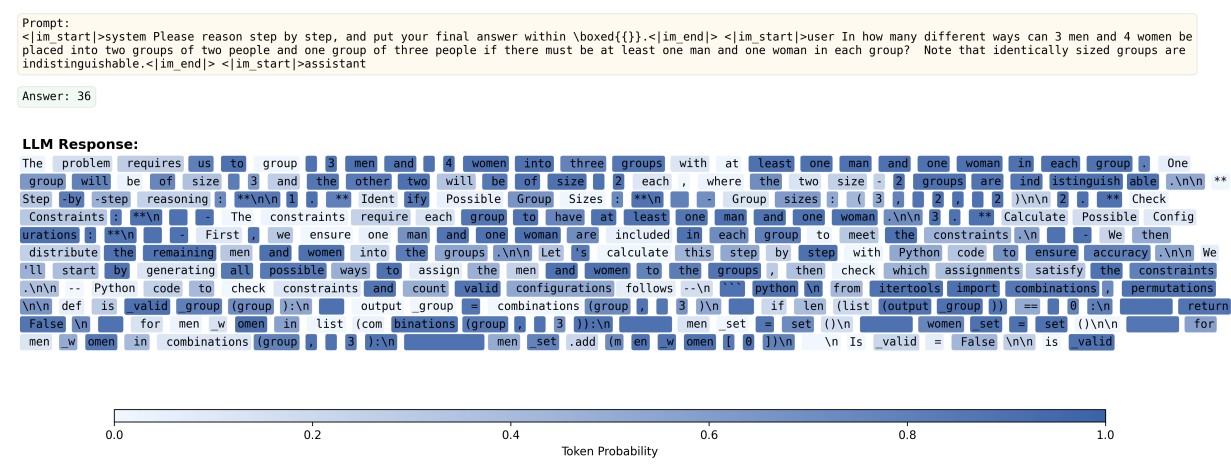

*Figure 4.* Additional Case 1: The model repeatedly restates the problem setup and constraints as long high-probability blocks, inflating CoT length with little new reasoning content.

---

[1] https://github.com/QwenLM/Qwen2.5-Math

**High-probability verification-style tail.** As shown in Fig 5. The prompt is a straightforward unit-conversion and scaling problem, whose computation reduces to $0.25 \times 22000 = 5500$. Although the model returns the correct answer, the trace contains an extended, template-like explanation and an appended pseudo-`Python` verification block (defining constants, converting units, and printing the result). The token-probability visualization highlights this verbose tail as a long contiguous high-probability segment, suggesting near-deterministic generation with limited informational value. This case exemplifies how overthinking can manifest as high-probability redundant segments that substantially increase length while contributing little to correctness.

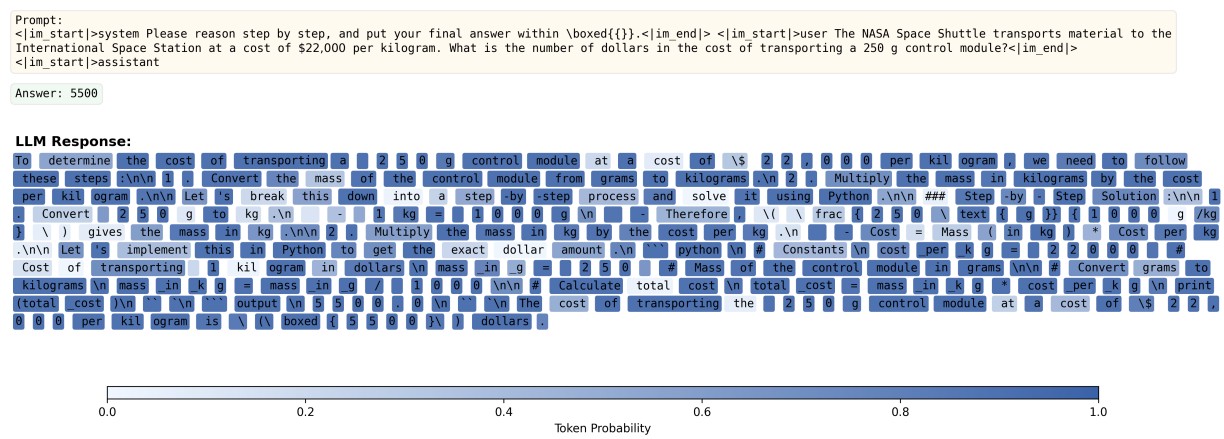

*Figure 5.* Additional Case 2: The model outputs the correct answer but generates a long high-probability tail (template-style explanation and pseudo-`Python` verification), illustrating a redundant segment that increases CoT length with little benefit to correctness.

## D.2. Detailed Results and training objective in Section 4.2.2

Below we summarize the implementations of different length objectives used in Section 4.2.2. These baselines differ only in the reward shaping term, and all other training and evaluation settings are kept identical to SLAT. Following the notation in the main text, $x$ denotes the input question and $L(z)$ denotes the CoT length.

`Truncation:`

$$r_{\text{trunc}} = r_{\text{ori}} \cdot \mathbb{I}(L(z) < L_0), \tag{27}$$

where $L_0$ is a predefined truncation length. We use $L_0 = 1024$ in our experiments.

`Length-MaxMin:`

$$r_{\text{MaxMin}} = r_{\text{ori}} + \lambda \cdot \begin{cases} \gamma, & \text{if correct,} \\ \min(0, \gamma), & \text{otherwise,} \end{cases} \tag{28}$$

where

$$\gamma = 0.5 - \frac{L(z) - L_{\min}}{L_{\max} - L_{\min}}, \tag{29}$$

and $L_{\max}$ and $L_{\min}$ denote the maximum and minimum response lengths within the GRPO sampling group, respectively.

`Length-Mean:`

$$r_{\text{Mean}} = r_{\text{ori}} - \lambda \cdot \sigma\left(\frac{L(z) - L_{\text{mean}}}{L_{\text{std}}}\right), \tag{30}$$

where $\sigma(\cdot)$ denotes the sigmoid function, and $L_{\text{mean}}$ and $L_{\text{std}}$ are the mean and standard deviation of the response lengths within the GRPO sampling group, respectively.

Here, $\lambda$ controls the strength of the length-penalty reward. For `Length-MaxMin` and `Length-Mean`, due to limited computational resources, we evaluate two candidate values of $\lambda$ $(0.25, 0.1)$ for each objective and report the best-performing results (one dominates the other).

We also provide the detailed numerical results for the experiments in Section 4.2.2. As shown in Table 5, SLAT achieves a consistently favorable accuracy-length trade-off compared to these token-uniform length objectives, improving efficiency while maintaining competitive accuracy.

*Table 5.* Comparison of models training the base model `Qwen2.5-Math-7B` with different objective. Accuracy (Acc.) and average token length (Len.) are reported for each dataset.

| | MATH 500 | | Olympiad Bench | | AMC23 | | AIME24 | | AIME25 | | Avg. | |
|---|---|---|---|---|---|---|---|---|---|---|---|---|
| | Acc. | Len. | Acc. | Len. | Acc. | Len. | Acc. | Len. | Acc. | Len. | Acc.↑ | Len.↓ |
| Qwen2.5-Math-7B | 42.8 | 1239 | 13.3 | 1602 | 30.7 | 1383 | 10.7 | 1620 | 4.2 | 1588 | 20.3 | 1486 |
| + GRPO | 82.2 | 688 | 42.9 | 999 | 69.3 | 927 | 33.8 | 1332 | 15.7 | 1325 | 48.8 | 1054 |
| + Truncation | 79.6 | 635 | 40.7 | 868 | 68.8 | 879 | 30.2 | 1097 | 11.5 | 1140 | 46.2 | 923 |
| + Length-MaxMin | 79.9 | 436 | 41.3 | 583 | 69.2 | 438 | 28.5 | 783 | 9.8 | 847 | 45.7 | 617 |
| + Length-Mean | 77.4 | 347 | 38.3 | 470 | 67.0 | 436 | 27.9 | 642 | 9.1 | 682 | 43.9 | 515 |
| + SLAT$_{w=5,\lambda=0.001}$ | 78.0 | 320 | 38.7 | 441 | 68.1 | 429 | 25.2 | 610 | 9.2 | 630 | 43.8 | 486 |
| + SLAT$_{w=7,\lambda=0.001}$ | 79.6 | 379 | 41.1 | 530 | 69.1 | 472 | 30.4 | 660 | 10.9 | 789 | 46.2 | 566 |
| + SLAT$_{w=7,\lambda=0.005}$ | 77.6 | 374 | 38.8 | 494 | 67.4 | 426 | 28.2 | 551 | 8.0 | 649 | 44.0 | 498 |
| + SLAT$_{w=9,\lambda=0.001}$ | 80.3 | 442 | 41.9 | 619 | 69.2 | 597 | 31.0 | 1161 | 12.0 | 1000 | 46.9 | 763 |

## D.3. Results on other domains

*Table 6.* Comparison of models trained from the base model `DeepSeek-R1-Distill-Qwen-7B` (denoted as Original). Accuracy (Acc.) and average token length (Len.) are reported for each dataset. For baselines, we directly evaluate their publicly released checkpoints under the same evaluation protocol. The last row additionally reports the relative change (%) of SLAT with respect to the original model.

| | MMLU | | GPQA | |
|---|---|---|---|---|
| | Acc.↑ | Len.↓ | Acc.↑ | Len.↓ |
| Original | 53.2 | 1674 | 47.5 | 6663 |
| LC-R1 | 58.0 | 880 | 49.5 | 4249 |
| DAST | **58.3** | 1612 | **51.0** | 6273 |
| TLMRE | 50.8 | 1023 | 47.9 | 5255 |
| L1-Max | 55.3 | 973 | 47.0 | **2342** |
| SLAT | **58.3** | **756** | 50.6 | 3234 |
| | +9.6% | -55% | +6.5% | -51% |

## D.4. Qualitative Case Analysis

We provide qualitative analysis results in Fig. 6 and Fig. 7.

Question: What is the sum of 2 and 3?

| DeepSeek-R1-Distill-Qwen-7B | DeepSeek-R1-Distill-Qwen-7B-SLAT |
|---|---|
| <think>I need to find the sum of the numbers 2 and 3. To do this, I will start by adding the two numbers together using basic addition principles. First, I'll add 2 and 3 to get 5. This is a straightforward addition of two single-digit numbers. To verify the result, I can use the commutative property of addition, which states that the order of the addends does not affect the sum. So, adding 3 and 2 also gives me 5. Therefore, the sum of 2 and 3 is 5. </think> To find the sum of **2** and **3**, follow these simple steps: 1. **Start with the first number:** \\[ 2 \\] 2. **Add the second number:** \\[ 2 + 3 \\] 3. **Calculate the addition:** \\[ 2 + 3 = 5 \\] 4. **Final Answer:** \\[ \\boxed{5} \\] | <think> First, I identify the two numbers involved in the addition: 2 and 3. Next, I add the first number to the second number: 2 plus 3 equals 5. Therefore, the sum of 2 and 3 is 5. </think> **Solution:** We need to find the sum of 2 and 3. \\[ 2 + 3 = \\boxed{5} \\] |

*Figure 6.* Qualitative comparison of question $2 + 3$. Given a simple question, the baseline distilled reasoner produces an overlong CoT with redundant scaffolding (restating the task, invoking generic principles, and listing trivial steps) despite quickly reaching the correct answer. The SLAT-trained model returns the same correct result with a substantially shorter and more focused trace.

Question:
Mikka wants to order a pizza with two different toppings. He has 8 different toppings to choose from. How many different pizzas could he order?

| DeepSeek-R1-Distill-Qwen-7B | DeepSeek-R1-Distill-Qwen-7B-SLAT |
|---|---|
| <think> Okay, so Mikka wants to order a pizza with two different toppings, and he has 8 different toppings to choose from. Hmm, I need to figure out how many different pizzas he could order. Let me think about this step by step.\n\nFirst, when the problem says he wants a pizza with two different toppings, I believe this is a combination problem because the order of the toppings doesn't matter on a pizza. Whether he puts tomato first or cheese first, it's still the same pizza. So, combinations are used when the order doesn't matter, right?\n\nSo, Mikka has 8 toppings to choose from, and he wants to choose 2 of them. The formula for combinations is C(n, k) = n! / (k!(n - k)!), where n is the total number of items, and k is the number of items to choose. In this case, n is 8 and k is 2.\n\nLet me write that out: C(8, 2) = 8! / (2!(8 - 2)!).\n\nSimplifying the denominator first, (8 - 2)! is 6!, so the formula becomes C(8, 2) = 8! / (2! * 6!). \n\nNow, I remember that factorials can be simplified by expanding them. 8! is 8 × 7 × 6 × 5 × 4 × 3 × 2 × 1, but since we have 6! in the denominator, those terms will cancel out. So, let's see:\n\n8! / 6! = (8 × 7 × 6!) / 6! = 8 × 7 = 56.\n\nSo now, the equation is C(8, 2) = 56 / (2!). \n\nWhat's 2! equal to? Well, that's just 2 × 1 = 2.\n\nSo, plugging that in, we get C(8, 2) = 56 / 2 = 28.\n\nTherefore, there are 28 different ways Mikka can choose 2 different toppings from 8.\n\nWait, let me double-check that to make sure I didn't make a mistake. So, another way to think about it is, for the first topping, Mikka has 8 choices. For the second topping, he has 7 choices because he can't choose the same topping again. So that would be 8 × 7 = 56.\n\nBut hold on, since the order doesn't matter, choosing topping A and then topping B is the same as choosing topping B and then topping A. So, we have to divide by the number of ways the two toppings can be arranged, which is 2! = 2. So, 56 / 2 = 28. Yeah, that matches what I got before. So, that seems consistent.\n\nAlternatively, I can think of this as pairings. With 8 toppings, how many unique pairs can we form? The first topping pairs with 7 others, the second with 6, and so on, but again, since each pair is counted twice, you have to divide by 2. So, (8 × 7) / 2 = 28. Yeah, same result.\n\nI think that's solid. So, Mikka has 28 different possible pizzas he could order with two different toppings. I don't see any mistakes in the reasoning. So, I'm confident with that answer.\n\n**Final Answer**\nThe number of different pizzas Mikka could order is \\boxed{28}.\n</think>\n\nMikka wants to order a pizza with two different toppings from 8 available options. This is a combination problem because the order of toppings does not matter.\n\nThe formula for combinations is given by:\n\\[ C(n, k) = \\frac{n!}{k!(n - k)!} \\]\n\nHere, \\( n = 8 \\) and \\( k = 2 \\). Plugging in these values:\n\\[ C(8, 2) = \\frac{8!}{2!(8 - 2)!} \\]\n\nSimplifying the factorials:\n\\[ C(8, 2) = \\frac{8 \\times 7 \\times 6!}{2 \\times 1 \\times 6!} \\]\n\nThe \\( 6! \\) terms cancel out:\n\\[ C(8, 2) = \\frac{8 \\times 7}{2 \\times 1} = \\frac{56}{2} = 28 \\]\n\nThus, the number of different pizzas Mikka could order is \\boxed{28}." | <think> He needs to choose 2 different toppings out of 8 available options.\n\nThe number of ways to choose 2 toppings from 8 is calculated using the combination formula, which is denoted as C(8, 2).\n\nUsing the combination formula:\nC(8, 2) = 8! / (2! * (8 - 2)!) = (8 * 7) / (2 * 1) = 28</think>\n\nTo determine how many different pizzas Mikka can order with two different toppings from a selection of 8, we can use the concept of combinations.\n\n**Step 1:** Mikka needs to choose 2 different toppings out of 8 available.\n\n**Step 2:** The number of ways to choose 2 toppings from 8 is calculated using the combination formula:\n\n\\[ \\text{Number of combinations} = \\binom{8}{2} = \\frac{8!}{2!(8-2)!} = \\frac{8 \\times 7}{2 \\times 1} = 28\\]\n\n**Final Answer:**\n\n\\[\\boxed{28}\\] |

*Figure 7.* Qualitative comparison of a question in MATH500. Both models produce the correct answer, but the original distilled reasoner continues with redundant scaffolding (repeatedly restating the combination setup and re-deriving the same quantity via multiple equivalent explanations), whereas SLAT gives a concise derivation and stops once the answer is determined.

