# OpenReview forum: "SLAT: Segment-Level Adaptive Trimming for Efficient CoT Reasoning"
_ICML.cc/2026/Conference — ICML 2026 regular_

### Official Review · Reviewer_iQFR · 2026-03-02

**Soundness:** 3
**Presentation:** 3
**Significance:** 3
**Originality:** 3
**Overall Recommendation:** 4
**Confidence:** 3

**Summary:**

This paper presents SLAT, an RL-based framework aimed at mitigating the "overthinking" phenomenon in Large Reasoning Models where excessively long CoT traces incur high computational costs without corresponding gains in accuracy. The authors observe that redundancy typically clusters in high-probability, near-deterministic segments—such as repetitive prompt restatements or trivial deductions—and they derive a theoretical characterization of suboptimality within a correctness-length trade-off objective. To operationalize this insight, SLAT employs a sliding-window proxy during reinforcement learning, specifically integrated into the GRPO algorithm, to detect and selectively penalize these long, redundant stretches within correct reasoning traces. By focusing on localized, segment-aware suppression rather than applying uniform token-level penalties, the method prunes verbose scaffolding while preserving the essential logical structures required for complex tasks. Experimental results across various mathematical and STEM benchmarks indicate that SLAT can reduce the average reasoning length of strong distilled and base models by approximately 50% while maintaining competitive accuracy, effectively establishing a superior accuracy-efficiency Pareto frontier compared to standard length-regularization baselines.

**Compliance With Llm Reviewing Policy:**

Affirmed.

**Final Justification:**

I believe this work is interesting and meaningful, and the authors' rebuttal has adequately addressed my concerns.

**Key Questions For Authors:**

1. How was the joint probability threshold ($\tau=0.9$) for the sliding window determined? Are there corresponding ablation studies to demonstrate the rationality and optimality of this specific value?


2. Could the proposed mechanism misidentify and penalize necessary, long, high-probability segments essential for complex reasoning—such as the full citation of theorems, detailed formula expansions, or formal definitions?


3. Does this trimming mechanism risk inducing "reward hacking" during reinforcement learning? Specifically, could the model learn to bypass penalties by replacing standard, natural reasoning with obscure expressions, rare vocabulary, or high-entropy "filler" to avoid high-probability segments? Would this compromise the rigor and consistency of the model’s internal logic?

**Limitations:**

1.  The authors should explicitly discuss the sensitivity of the fixed sliding-window strategy to the hyperparameters $w$ (window size) and $\tau$ (probability threshold), and provide empirical analysis on how variations in these parameters affect the robustness of the accuracy-efficiency trade-off.


2. There should be a discussion regarding whether, and to what extent, the segment-level reinforcement learning process impacts the model’s internal logical consistency and the overall structural integrity of its reasoning chains.


3. I suggest the authors explore the potential risk of "reward hacking," where the model might learn to circumvent penalties for high-probability segments by intentionally generating "low-probability" yet low-quality or linguistically unnatural reasoning steps, thereby undermining the goal of achieving concise yet faithful reasoning.

**Strengths And Weaknesses:**

### Soundness

* **Strengths**: The study is built upon a solid theoretical foundation, establishing a rigorous mathematical framework to derive sufficient conditions for segment suboptimality in CoT reasoning. This theoretical characterization aligns closely with extensive empirical results. The authors provide thorough validation across multiple mathematical reasoning benchmarks, demonstrating that the SLAT framework significantly reduces reasoning length by approximately 50% without compromising accuracy, effectively establishing a superior accuracy-efficiency Pareto frontier.


* **Weaknesses**: There is a "proxy gap" between the theoretical derivation and the actual engineering implementation . The sliding-window counting method serves as a heuristic proxy for the complex theoretical conditions, yet the paper lacks a detailed ablation study on the core hyperparameter $\tau$ (joint probability threshold) to justify its optimality . Furthermore, the paper does not deeply explore the potential risk of "reward hacking" in reinforcement learning—specifically, whether the model might learn to circumvent length penalties by generating low-probability, obscure expressions, which could inadvertently compromise the naturalness and coherence of its logic.



### Presentation

* **Strengths**: The paper is well-structured and logically clear, following a coherent narrative that begins with identifying redundancy in current reasoning models, moves to theoretical characterization, and concludes with the implementation of the RL training framework . This flow makes the overall methodology easy to follow. Additionally, the high-quality visual aids intuitively demonstrate the behavioral shifts in the model before and after training, significantly enhancing the work's readability.



### Significance

* **Strengths**: The research accurately addresses the highly relevant and practical challenge of "overthinking" in modern large reasoning models. The efficient reasoning capabilities demonstrated by SLAT hold substantial practical utility for reducing inference costs in industrial deployments and enhancing training throughput, suggesting strong potential for future research directions .


* **Weaknesses**: The impact of the experimental results is primarily concentrated within the domain of mathematical reasoning. While the authors show some cross-domain generalization on MMLU-STEM and GPQA, the generalizability of the proposed method to non-deterministic or more divergent complex reasoning tasks requires further large-scale empirical support .



### Originality

* **Strengths**: The paper introduces a novel "segment-level" perspective, moving beyond the coarse-grained, token-uniform length penalties prevalent in existing literature. By being the first to derive mathematical suboptimality conditions for reasoning segments under an explicit accuracy-efficiency trade-off, the work provides an original theoretical lens for the study of efficient Chain-of-Thought reasoning.

---

> ### Author Rebuttal · Authors · 2026-03-31
>
> We thank the reviewer for the careful and encouraging feedback. We are encouraged that the reviewer recognizes the sound theoretical foundation of the work, the clear presentation and overall readability, the practical significance of addressing overthinking in large reasoning models, and the originality of the segment-level perspective beyond token-uniform length control. We address the main concerns below. An anonymous link to the experiment is provided at https://postimg.cc/47M0JQnF.
>
> **Soundness_Weaknessess**:
>
> Regarding the “proxy gap”, our main intent is that C(z) serves as a tractable surrogate, while Proposition 3.1 provides the underlying theoretical insight for why trimming long, high-probability segments can improve the correctness-length objective. We do not claim that the sliding-window count is a calibrated estimator of the full sufficient condition. Instead, the theory motivates the direction of the surrogate, and we will clarify this distinction more explicitly in the revision.
>
> As for the concerns about the $\tau$ ablation and potential reward hacking, we address these points in detail in our later responses.
>
> **Generability (Significance_Weakness)**:
>
> We agree that our strongest evidence is currently concentrated on mathematical reasoning, which is also the main focus of many recent RL-for-LLM reasoning works (LC-R1, AdaptThink, DAST, TLMRE, and L1-Max). **To partially address whether SLAT also transfers to less formulaic, non-mathematical multi-step reasoning, due to limited resources, we additionally ran a logical reasoning experiment**.
> Specifically, we applied SLAT training to Llama-3.2-3B-Instruct on the K&K dataset, which has been used in recent LLM reasoning studies (e.g. Logic-RL). We trained on 3-person scenarios and evaluated on both 3-person (-3ppl) and 4-person (-4ppl) tasks. As shown in Table 2, SLAT reduces average response length by 19% (-3ppl) and 24% (-4ppl) while maintaining accuracy comparable to GRPO.
> This shows that the benefit of segment-level trimming is not limited to formulaic math reasoning and can extend to non-quantitative, multi-step logical reasoning domains.
>
> **Ablation (Q1&L1)**:
>
> We agree that $\tau$ deserves clearer analysis. In the revised ablation, we add three extra runs on top of the original table, including $\tau=0.8$ and $\tau=0.95$ under the default setting ($w=7, \lambda=0.001$). As shown in **Table 3**, the results show the expected trade-off: smaller $\tau$ makes trimming more aggressive, while larger $\tau$ makes it more conservative. In our setting, $\tau=0.9$ is not claimed to be universally optimal, but it gives a good accuracy-compression balance.
> **We also have ablation and similar findings about $\lambda$ and $w$**.
> In practice, SLAT behaves much like a **one-knob method**. We find that fixing $\tau$ and $\lambda$ and tuning only w already provides a strong and smooth trade-off, so w can be treated as the main control of trimming aggressiveness in practice.
>
> **Discussion on segment-level method (Q2&L2)**:
>
> We agree that an important question is whether SLAT may occasionally penalize necessary long high-probability segments and whether this could affect the overall structural integrity of the reasoning chain. Our current results suggest that such failures are not dominant, but they can occur. In addition to the qualitative case studies already shown in the paper, we conducted a preliminary manual inspection on 20 MATH500 samples generated by Qwen2.5-Math-7B + SLAT. We grouped them into (1) clear and concise reasoning, (2) correct reasoning with possible redundancy, and (3) incorrect or skipped-step reasoning, **with counts 13 / 4 / 3, respectively**.
> Most SLAT outputs remain logically sound and reasonably compact, while a smaller fraction still exhibit redundancy or structural reasoning errors. We agree that this issue deserves a more explicit discussion, and in the revision we will expand this qualitative-case analysis.
>
> **Reward hacking (Q3&L3)**:
>
> We agree that potential reward hacking is an important issue to consider. Importantly, SLAT is not optimized in isolation: the trimming term is combined with the original RL reward, which still encourages the model to preserve task-correct, meaningful, and coherent reasoning needed for correct answers.
>
> Our preliminary qualitative analysis also does not suggest that this behavior is dominant. As noted in the previous response. While this does not rule out all reward-hacking behaviors, it suggests that SLAT does not primarily drive the model toward unnatural or incoherent reasoning in practice.
>
> We agree that this is an important direction for deeper analysis, and we will discuss this limitation more explicitly and further examine such failure modes in future work.
>
> Thanks for your feedback. We hope these revisions address your concerns. We are happy to provide further clarifications to facilitate the review process.

---

> > ### Author Rebuttal · Reviewer_iQFR · 2026-04-03
> >
> > Thank you for the rebuttal. I find this work to be a meaningful contribution. The additional analyses provided by the authors will help future readers better understand and contextualize the paper. I will keep my score unchanged.

---

> > > ### Author Response · Authors · 2026-04-04
> > >
> > > Thank you very much for your thoughtful reviews. Your feedback has significantly improved the quality of our paper. We also greatly enjoyed the rebuttal process and the opportunity to engage with your comments.

---

### Official Review · Reviewer_9sw3 · 2026-03-09

**Soundness:** 3
**Presentation:** 3
**Significance:** 3
**Originality:** 3
**Overall Recommendation:** 4
**Confidence:** 3

**Summary:**

In this work, the primary challenge addressed is the excessive length of reasoning chains in large reasoning models.
Unlike prior approaches that penalize every token using a length-penalty reward (e.g., applying a negative reward if the total reasoning length exceeds a certain threshold), the authors propose penalizing only the segments that exhibit high entropy. The core motivation is that high-entropy segments typically do not contribute to the progression of the reasoning process.
The proposed method, SLAT, counts the number of these high-entropy segments and applies a penalty reward proportional to that count. In experiments across several mathematical reasoning benchmarks, SLAT demonstrated a length reduction of approximately 50% compared to the original model while maintaining the same level of accuracy.

**Compliance With Llm Reviewing Policy:**

Affirmed.

**Final Justification:**

I have read the authors' rebuttal and find that it adequately addresses the concerns raised in my initial review. I am satisfied with the clarifications provided and maintain my positive assessment of the paper.

**Key Questions For Authors:**

1. Could you provide a more detailed description of how each baseline method works and how it concretely differs from SLAT (e.g., token-level vs. segment-level, reward design)? This would significantly improve the interpretability of the experimental results. Without this, it is difficult to assess what specifically drives SLAT's improvements.
2. Have you analyzed how many redundant segments are missed due to the fixed window size? Have you considered adaptive window approaches? Understanding the coverage of the current detection mechanism would help assess the robustness of the method.
3. Are there failure cases where SLAT suppresses necessary reasoning steps, leading to incorrect answers? An analysis of such cases would help clarify the boundary between beneficial trimming and harmful suppression, and would strengthen confidence in the method's reliability.

**Limitations:**

The paper already stated its limitation in Section 6.

**Strengths And Weaknesses:**

### Strengths

- The rationale behind the method is well-motivated and clearly articulated in Section 3. The progression from the theoretical characterization of segment suboptimality (Proposition 3.1) to the practical implementation via sliding-window reward shaping is logical and convincing. The idea of targeting high-probability segments rather than applying uniform length penalties is a principled and novel contribution.
- The experimental evaluation is extensive, covering multiple models, numerous baselines, and several mathematical reasoning benchmarks. The inclusion of out-of-domain evaluation further strengthens the empirical validation.
- The paper is clearly written and easy to follow overall. The structure from motivation to theory to method to experiments is well-organized and makes the contribution accessible.

### Weaknesses

- The paper lacks a concrete description of how each baseline method works and how it differs from SLAT. While many baselines are listed and cited, their mechanisms (e.g., what reward they use, whether they operate at token-level or segment-level) are not explained. This makes it difficult to fully interpret the experimental comparisons and understand why SLAT outperforms them.
- The sliding window approach relies on a fixed window size $w$ and threshold $\tau$, which introduces sensitivity to these hyperparameters. A hard window boundary may miss redundant segments that are slightly shorter than w or have probabilities just below $\tau$. While the authors fix $\tau = 0.9$ and vary $w$ in practice, this rigidity could limit the method's adaptability across different domains or models.

#### Minor Comments
- In Figure 1, the current visualization only shows token probability via color shading, but it does not clearly convey the core motivation that high-probability segments are low-information and redundant. It would be more effective to redesign the figure to explicitly highlight why these segments are unnecessary (e.g., annotating the content of each segment to show repetitiveness).
- In Figure 2, the background coloring hinders readability and should be removed. The legend should be enlarged, and the meaning of the green markers (whether they represent baselines) should be explicitly labeled.
- The current $C(z)$ counts all high-probability windows equally regardless of their position in the reasoning trace. It may be worth considering a position-aware variant that applies stronger penalties to high-probability segments appearing later in the trace, where redundancy is more likely, while being more lenient toward earlier segments that may serve as foundational reasoning steps.

---

> ### Author Rebuttal · Authors · 2026-03-31
>
> We thank the reviewer for the careful reading and thoughtful feedback, including many helpful suggestions on both the technical details and presentation. We are encouraged that the reviewer finds the motivation and overall rationale of the method clear, and that the connection from the theoretical characterization in Proposition 3.1 to the practical sliding-window reward shaping is logical and convincing. We also appreciate the recognition of the experimental breadth. We address the main concerns below. An anonymous link to the experiment is provided at https://postimg.cc/47M0JQnF.
>
> **More Discussion about baseline (W1&Q1)**:
>
> We discuss the baselines in the related work section, but we agree that their mechanisms can be explained more clearly to better contextualize the comparisons. Specifically, LC-R1 combines a global Length Reward with a Compress Reward to suppress repeated post-solution checking; AdaptThink learns an instance-level binary Thinking/NoThinking decision; DAST is also instance-adaptive, using a difficulty-estimated Token Length Budget together with budget-aware shaping and preference optimization; TLMRE applies a global length penalty with α controlling its strength; and L1-Max performs explicit budget control via LCPO under a maximum-length constraint. In the revision, we will present these baselines more clearly and include **a compact comparison table summarizing their reward design, granularity of control (token / instance / segment), and key training mechanism**.
>
> **About $\tau$ and fixed window  (W2&Q2)**:
>
> We agree that $\tau$ deserves clearer analysis. In the revised ablation, we add three extra runs on top of the original table, including $\tau=0.8$ and $\tau=0.95$ under the default setting ($w=7, \lambda=0.001$). As shown in **Table 3**, the results show the expected trade-off: smaller $\tau$ makes trimming more aggressive, while larger $\tau$ makes it more conservative. In our setting, $\tau=0.9$ is not claimed to be universally optimal, but it gives a good accuracy-compression balance.
> **We also have ablation and similar findings about $\lambda$ and $w$**.
> In practice, SLAT behaves much like a **one-knob method**. We find that fixing $\tau$ and $\lambda$ and tuning only w already provides a strong and smooth trade-off, so w can be treated as the main control of trimming aggressiveness in practice.
>
> Thanks for the helpful point on fixed window. We would like to say that the sliding-window count is not a single hard match: because overlapping windows are counted, its value grows roughly proportionally to the length of a sustained high-probability stretch. In this sense, the mechanism is less brittle than a one-shot detector with a fixed boundary. We agree that adaptive-window variants are a promising extension, and we will clarify this limitation and discuss such variants in the revision.
>
> **Suggestion on Figures (W3&W4)**:
>
> Thank you for the careful feedback! In the revision, we will redesign Figure 1 to more explicitly show why the highlighted high-probability segments are redundant, by annotating representative repetitive and low-information spans. We will also improve the readability of Figure 2.
>
> **Position aware method (W5)**:
>
> We agree that a position-aware variant is worth exploring, and we will add this discussion in the revision. Our current design is intentionally position-agnostic to keep SLAT simple and plug-and-play, without introducing extra assumptions about where redundancy occurs. We also note that position alone is not always a reliable proxy: early spans can still be boilerplate or restatement, while later spans may contain indispensable bridging steps. A position-aware weighting could therefore be a promising extension, and we leave it as future work.
>
> **Qualitative Analysis / bad case (Q3)**:
>
> Thank you for this valuable suggestion. We conduct a preliminary manual inspection on 20 MATH500 samples generated by Qwen2.5-Math-7B + SLAT. We categorized them into three groups: (1) clear and concise reasoning, (2) correct reasoning with possible redundancy, and (3) incorrect or skipped-step reasoning. **The current counts are 13 / 4 / 3, respectively**. Most SLAT outputs remain logically sound and reasonably compact, while a smaller fraction still exhibit redundancy or reasoning errors. We agree that a broader failure-case analysis would make the empirical story more balanced and credible. We will expand this manual study and include the results in the revision.
>
> Thanks for your feedback. We hope these revisions adequately address your concerns. We are happy to provide further clarifications to facilitate the review process.

---

> > ### Author Rebuttal · Reviewer_9sw3 · 2026-04-02
> >
> > I appreciate the authors for preparing a strong response to my review. Adding the baseline comparison table will be very helpful in guiding readers to understand the unique advantages of SLAT compared to existing methods. I will maintain my positive score.

---

> > > ### Author Response · Authors · 2026-04-04
> > >
> > > Thank you very much for your careful reviews. Your feedback has significantly improved the quality of our paper. We also greatly enjoyed the rebuttal process and the opportunity to engage with your comments.

---

### Official Review · Reviewer_drmm · 2026-03-12

**Soundness:** 2
**Presentation:** 3
**Significance:** 3
**Originality:** 3
**Overall Recommendation:** 4
**Confidence:** 4

**Summary:**

This paper addresses the efficiency of Chain-of-Thought (CoT) reasoning in Large Reasoning Models (LRMs). The authors argue that standard length-penalties are too coarse (token-uniform) and propose SLAT, a segment-level reward-shaping method. SLAT identifies and penalizes long, high-probability segments during RL training, hypothesizing they represent low-information redundancy. Evaluated on DeepSeek-R1-Distill and Qwen base models, SLAT demonstrates a significant reduction in reasoning length (up to 50%) while maintaining or slightly improving accuracy across math and STEM benchmarks (MATH500, AIME, etc.).

**Compliance With Llm Reviewing Policy:**

Affirmed.

**Final Justification:**

I thank the authors for their clear and constructive rebuttal. The responses and additional experiments address several of my earlier concerns.

First, the authors now report multi-seed results and standard errors for their main SLAT setting. Although the number of seeds is limited, the variance appears reasonably small and supports the stability of the main trend.

Second, the additional logical reasoning experiment (K&K dataset) helps address my concern about generalization beyond mathematical reasoning. The results show that SLAT can reduce response length by about 19–24% while maintaining comparable accuracy, suggesting that the approach may transfer to other multi-step reasoning tasks.

Third, the rebuttal provides additional ablations on τ, clarifying its role in controlling trimming aggressiveness and improving the understanding of the method.

Finally, the preliminary qualitative analysis of generated samples provides useful insight into the behavior of SLAT and suggests that most outputs remain logically sound while becoming more concise.

Overall, while some limitations remain (e.g., evaluation breadth and limited qualitative analysis), the authors have made a good effort to address the main concerns. Given the simplicity, practicality, and potential usefulness of the method for improving the accuracy–efficiency trade-off in reasoning RL pipelines, I am increasing my score from 3 to 4.

**Key Questions For Authors:**

# Questions
1. Can you provide variance/confidence intervals across multiple RL seeds for the main results (Table 1 & 2)?
2. How sensitive is the accuracy-length trade-off to the interaction between window size $w$ and threshold $\tau$ ? Is $\tau=0.9$ truly optimal across different model scales?
3. Have you performed a qualitative or human study to check if any penalized "high-probability" segments actually contained indispensable logical steps?
4. Does the method maintain its effectiveness on tasks where reasoning is less formulaic and repetition is not the primary source of length (e.g., long-form legal or commonsense reasoning)?

**Limitations:**

1. Failure-case analysis would make the empirical story more balanced and credible.
2. The paper would be substantially strengthened by adding seed variance and significance analyses, ablations over $w, \tau, \lambda$, and more direct evidence that high-probability windows correlate with removable redundancy rather than merely easy tokens.

**Strengths And Weaknesses:**

# Strengths
1. Tackles the "overthinking" and repetitive trace issues common in RL-trained reasoning models, moving beyond simple sequence-level length constraints.
2. Achieves impressive compression (e.g., 50% token reduction on R1-Distill-7B) with minimal accuracy trade-offs, outperforming several recent baselines like LC-R1 and AdaptThink.
3. The sliding-window high-probability count is easy to implement and integrate into existing GRPO-style pipelines.
# Weaknesses
1. The theoretical "sufficient condition" for suboptimality is loosely bridged to the practical "high-probability sliding window" detector. It remains unproven whether this detector accurately isolates redundancy or merely identifies easy-to-predict, yet logically necessary, steps.
2. The accuracy gains are relatively small and validated on limited datasets (e.g., AIME), without reporting seed variance or conducting significance tests.
3. While cross-domain tests are included, the strongest evidence is concentrated in math. It is unclear if the redundancy patterns SLAT targets exist or behave similarly in non-quantitative, multi-step logical reasoning.

---

> ### Author Rebuttal · Authors · 2026-03-31
>
> We thank the reviewer for the thoughtful and constructive comments. We are encouraged that the reviewer recognizes SLAT as a simple yet effective approach to addressing overthinking and repetitive reasoning traces beyond sequence-level length control. We also appreciate the recognition of its strong compression performance with minimal accuracy trade-off, as well as its practical ease of integration into standard GRPO-style pipelines. We address the main concerns below. An anonymous link to the experiment is provided at https://postimg.cc/47M0JQnF.
>
> **Std (W2&Q1)**:
>
> Thank you for this important suggestion. We would also like to clarify that SLAT is **not primarily designed to maximize accuracy**, but to improve the accuracy–efficiency trade-off by reducing redundant reasoning with minimal performance loss. Due to limited training resources, we were only able to rerun our main SLAT setting with 3 RL seeds.  As shown in **Table 1**, the standard errors are not large across datasets, suggesting that the main trend is reasonably stable.
>
> We also note that many baselines in Table 1 are evaluated from publicly released checkpoints, so matched multi-seed retraining is not feasible on our side. We will therefore add seed variance for our main method and clarify this limitation more explicitly in the revision.
>
> **Generablity (W3&Q4&L2)**:
>
> We agree that our strongest evidence is currently concentrated on mathematical reasoning, which is also the main focus of many recent reasoning works (LC-R1, AdaptThink, DAST, TLMRE). To address whether SLAT also transfers to less formulaic, non-mathematical multi-step reasoning, due to limited resources, we additionally ran a logical reasoning experiment.
> Specifically, we applied SLAT training to Llama-3.2-3B-Instruct on the K&K dataset, which has been used in recent LLM reasoning studies (e.g. Logic-RL). We trained on 3-person scenarios and evaluated on both 3-person (-3ppl) and 4-person (-4ppl) tasks. As shown in **Table 2**, SLAT reduces average response length by 19% (-3ppl) and 24% (-4ppl) while maintaining accuracy comparable to GRPO.
> This shows that the benefit of segment-level trimming is not limited to formulaic math reasoning and can extend to non-quantitative, multi-step logical reasoning domains.
>
> **Ablation (Q2&L2)**:
>
> We agree that $\tau$ deserves clearer analysis. In the revised ablation, we add three extra runs on top of the original table, including $\tau=0.8$ and $\tau=0.95$ under the default setting ($w=7, \lambda=0.001$). As shown in **Table 3**, the results show the expected trade-off: smaller $\tau$ makes trimming more aggressive, while larger $\tau$ makes it more conservative. In our setting, $\tau=0.9$ is not claimed to be universally optimal, but it gives a good accuracy-compression balance.
> **We also have ablation and similar findings about $\lambda$ and $w$**.
> In practice, SLAT behaves much like a **one-knob method**. We find that fixing $\tau$ and $\lambda$ and tuning only w already provides a strong and smooth trade-off, so w can be treated as the main control of trimming aggressiveness in practice.
>
> **About Proposition(W1)**:
>
> Thank you for this important point. Our intent is that the high-probability sliding-window count serves as a tractable surrogate, while the sufficient condition provides the underlying theoretical insight for why trimming long, high-probability segments can improve the correctness–length objective. We do not claim that the detector is a calibrated test that perfectly separates removable redundancy from all logically necessary steps. Importantly, SLAT is not optimized in isolation: it is combined with the original RL reward, which still encourages the model to preserve task-correct and meaningful reasoning steps needed to reach the right answer. In this sense, the trimming term acts as an auxiliary bias against overly confident redundant spans, while the original reward anchors the policy toward maintaining necessary reasoning.
>
> **Qualitative Analysis / bad case (Q3&L1)**:
>
> Thank you for this valuable suggestion. We conduct a preliminary manual inspection on 20 MATH500 samples generated by Qwen2.5-Math-7B + SLAT. We categorized them into three groups: (1) clear and concise reasoning, (2) correct reasoning with possible redundancy, and (3) incorrect or skipped-step reasoning. **The current counts are 13 / 4 / 3**, respectively.
> Most SLAT outputs remain logically sound and reasonably compact, while a smaller fraction still exhibit redundancy or reasoning errors. We agree that a broader failure-case analysis would make the empirical story more balanced and credible. We will expand this manual study and include the results in the revision.
>
> Thanks for your feedback. We hope these revisions adequately address your concerns. We are happy to provide further clarifications to facilitate the review process.

---

> > ### Author Rebuttal · Reviewer_drmm · 2026-04-03
> >
> > I thank the authors for providing additional experimental results in response to my questions; having also reviewed the comments from the other reviewers, I am upgrading my overall recommendation score from 3 to 4.

---

> > > ### Author Response · Authors · 2026-04-04
> > >
> > > Thank you very much for your thoughtful reviews. Your feedback has significantly improved the quality of our paper. And we are glad that our responses have addressed your concerns.
> > >
> > > We also noticed your kind comment about raising the score, and we would be truly grateful if you could kindly update it in the system when convenient.

---

### Official Review · Reviewer_AKMm · 2026-03-15

**Soundness:** 3
**Presentation:** 3
**Significance:** 2
**Originality:** 3
**Overall Recommendation:** 4
**Confidence:** 3

**Summary:**

This paper addresses the "overthinking" problem in large reasoning models, where chain-of-thought (CoT) traces contain redundant, high-probability segments that inflate length without improving correctness. The authors derive a theoretical sufficient condition (Proposition 3.1) under which generating a particular segment is suboptimal under a correctness-length trade-off objective. Based on this, they propose SLAT, which uses a sliding-window mechanism to detect high-probability stretches in reasoning traces and penalizes them via reward shaping within the GRPO framework. Experiments on mathematical reasoning benchmarks show ~50% length reduction with competitive accuracy.

**Compliance With Llm Reviewing Policy:**

Affirmed.

**Key Questions For Authors:**

1. Have you measured how frequently the sufficient condition in Proposition 3.1 is actually satisfied for the windows flagged by C(z)?

2. What happens if you also apply the trimming penalty to incorrect rollouts (removing the correctness gate)?

3. How sensitive is performance to τ? Is 0.9 optimal, or simply "good enough"?

4. Can you provide results on at least one non-mathematical reasoning domain (e.g., code generation or logical puzzles) with training, not just evaluation transfer?

**Limitations:**

yes

**Strengths And Weaknesses:**

Strengths
1. Well-motivated problem with clear empirical grounding. The case studies in Figures 1, 3, and 4 effectively illustrate the overthinking phenomenon. The observation that redundancy concentrates in high-probability segments is intuitive and well-supported — high-probability token runs correspond to near-deterministic, low-information content that the model could essentially skip.
2. Clean theoretical contribution. Proposition 3.1 provides a principled justification for why long, high-probability segments are natural candidates for suppression. The proof is sound and the connection between the theoretical condition and the practical sliding-window proxy (Eq. 10) is clearly drawn. The "global viewpoint" discussion in Appendix B.1 is a nice touch, explaining why the segment-level view is more actionable than a whole-trace condition.
3. Strong empirical results on distilled models. Table 1 is compelling — SLAT achieves the best average accuracy (66.4) while cutting CoT length by 50% from the original model, Pareto-dominating most baselines. The comparison is reasonably fair, using official checkpoints under a unified evaluation protocol.
4. Simplicity of the method. Despite the theoretical framing, the actual mechanism (count sliding windows exceeding a probability threshold, penalize correct rollouts) is straightforward to implement. The authors' claim that it behaves like a "one knob" method is credible given the fixed τ=0.9 and small λ.

Weaknesses
1. Gap between theory and practice is larger than acknowledged. Proposition 3.1 provides a sufficient condition involving the full conditional correctness probability P_θ(A|x), the segment probability, and expected lengths — quantities that are not computed or estimated during training. The sliding-window count C(z) is described as a "proxy," but the paper doesn't rigorously characterize when this proxy faithfully captures the theoretical condition. The connection is argued informally (longer high-prob stretches produce more overlapping windows), but this is a monotonicity argument, not a calibration one. It would strengthen the paper to show empirically how often the theoretical condition is actually satisfied for the segments SLAT penalizes.
2. The correctness-gating design choice deserves more scrutiny. The trimming reward is applied only to correct rollouts (Eq. 11), with the justification that this prevents "degenerate shortcuts." However, this means SLAT provides no signal to shorten incorrect traces, which can also be extremely long and wasteful. More importantly, this design creates an asymmetry where the model receives a penalty for being correct-but-verbose but no penalty for being incorrect-and-verbose, which could subtly discourage the model from attempting harder reasoning paths. The paper doesn't discuss this trade-off or provide ablation on gating.
3. Limited task diversity. Nearly all results are on mathematical reasoning. While the MMLU-STEM and GPQA results in Table 2 gesture toward generalization, these are evaluated only on the distilled model setting with no training on those domains. The paper doesn't test on coding, logical reasoning, or open-ended generation tasks where the structure of "useful" vs. "redundant" reasoning may differ substantially. The claim about "efficient CoT reasoning" in the title is broader than what the experiments support.
4. Baseline comparisons in the base-model setting are incomplete. In Section 4.2.2, the baselines (Truncation, Length-MaxMin, Length-Mean) are all token-uniform length penalties — precisely the class of methods SLAT is designed to outperform. Missing are comparisons against other segment-aware or entropy-based methods (e.g., the step-entropy approach of Li et al. 2025, or DAST's difficulty-adaptive mechanism) in this controlled training setting. The distilled-model comparisons in Table 1 use released checkpoints that may differ in training data, epochs, and other confounds.
5. Sensitivity analysis is thin. The paper varies w and λ in Figure 2 but doesn't systematically study the interaction between these hyperparameters and the threshold τ. The claim that τ=0.9 works universally is not validated across different model families or domains. What happens at τ=0.8 or τ=0.95? Given that the method's effectiveness hinges on correctly identifying "high-probability" segments, this threshold is a critical design choice that warrants more investigation.
6. The sliding window treats all high-probability segments equally. A segment of high-probability tokens at the beginning of a trace (where the model is setting up the problem) may be quite different from one in the middle or tail. Position within the reasoning chain likely matters for whether a segment is truly redundant, but SLAT's penalty is position-agnostic. This seems like a meaningful limitation the paper should discuss.

---

> ### Author Rebuttal · Authors · 2026-03-31
>
> We thank the reviewer for the thoughtful and constructive comments. We are encouraged that the reviewer finds the problem well motivated, the case studies empirically convincing, the theoretical contribution principled and clean, and the recognition of SLAT’s strong empirical performance. We address the main concerns below. An anonymous link to the experiment is provided at https://postimg.cc/47M0JQnF .
>
> **Generability (W3&Q4 )**:
>
> Currently, we focus on mathematical reasoning, which aligns with many recent reasoning works (LC-R1,AdaptThink,DAST,TLMRE). We additionally ran a non-mathematical logical reasoning experiment. Specifically, we applied SLAT training to Llama-3.2-3B-Instruct on the K&K dataset, which has been used in recent LLM reasoning studies (e.g. Logic-RL). We trained on 3-person scenarios and evaluated on both 3-person (-3ppl) and 4-person (-4ppl) tasks. As shown in Table 2, **SLAT reduces average response length by 19% (-3ppl) and 24% (-4ppl) while maintaining accuracy comparable to GRPO**, suggesting the generability of SLAT.
>
> **Ablation (W5&Q3)**:
>
> We agree that $\tau$ deserves clearer analysis. In the revised ablation, we add three extra runs on top of the original table, including $\tau=0.8$ and $\tau=0.95$ under the default setting ($w=7, \lambda=0.001$). The results show the expected trade-off: smaller \tau makes trimming more aggressive, while larger \tau makes it more conservative. In our setting, $\tau=0.9$ is not claimed to be universally optimal, but it gives a good accuracy-compression balance.
> In practice, SLAT behaves much like a **one-knob method**. We find that fixing $\tau$ and $\lambda$ and tuning only w already provides a strong and smooth trade-off, so w can be treated as the main control of trimming aggressiveness in practice.
>
> **Proposition (W1&Q1)**:
>
> Our main intent is that C(z) serves as a **tractable surrogate**, while Proposition 3.1 provides the underlying **theoretical insight** for why trimming long, high-probability segments can be beneficial. Currently, we can not directly measure how often the sufficient condition is exactly satisfied for the windows flagged by C(z), since doing so would require post-hoc estimation of quantities such as the counterfactual conditional correctness and counterfactual expected length terms in Proposition 3.1, which are not available in standard training. **We agree that such an empirical analysis is an important and in future work we will explore reasonable estimators for these theoretical quantities**.
>
> **SLAT on positive sample (W2&Q2)**:
>
> Our current choice is intentionally conservative: for incorrect rollouts, excessive length may reflect either redundancy or genuinely unfinished reasoning on hard problems, so penalizing them uniformly could bias the model toward prematurely shortening difficult reasoning attempts. By restricting the trimming signal to correct trajectories, **SLAT only compresses reasoning paths that have already demonstrated task success**, while leaving unresolved trajectories less constrained. This intuition is **consistent with prior RL (e.g. Self-Imitation Learning [ICML] ) and RL-for-LLM work (e.g. Diversity-Aware Policy Optimization [NeurIPS])**, which suggests that additional learning signals are often anchored on successful or promising trajectories rather than uniformly imposed on all trajectories. Due to limited compute, we could not include this ablation during rebuttal but agree it is valuable and will add it in the future.
>
> **Position (W6)**:
>
> Yes, we agree that it is a useful limitation to clarify. Our current design is intentionally position-agnostic to keep SLAT simple and avoid hand-crafted assumptions about where redundancy occurs. **We also note that position alone may not always be a reliable proxy for redundancy**: early spans can be boilerplate or restatement, while later spans may still contain indispensable bridge steps. Empirically, the current design already achieves substantial compression with a small accuracy change.
>
> **Comparision (W4)**:
>
> For Table 1, our goal is to position SLAT against representative efficient-reasoning systems under a unified evaluation protocol, rather than to claim a fully controlled causal comparison. This is also common in recent LLM efficient reasoning work (e.g. Thinking-Free Policy Initialization, ATTNPO, ConPress). We also note that Step Entropy was not included in Table 1 because the author did not release the trained checkpoints.
>
> For Table 2, our goal was to isolate the effect of segment-level reward design under the same training pipeline. Under limited training resources, we prioritized baselines that can be implemented fairly in exactly the same pipeline. Step Entropy is not an apples-to-apples baseline for Table 2, since it uses a two-stage training pipeline. We agree that broader adaptive baselines, such as DAST would further strengthen the controlled comparison. However, due to limited resources, we leave it to future work.

---

> > ### Author Rebuttal · Reviewer_AKMm · 2026-04-04
> >
> > Overall, the rebuttal addresses several of our concerns partially but honestly. The core contribution — a simple, theoretically motivated method that achieves strong compression with minimal accuracy loss — remains solid. We maintain our score of weak accept, with the suggestion that the revision incorporate the K&K experiment, the $\tau$ ablation, and more careful framing of the theory-practice relationship.

---

> > > ### Author Response · Authors · 2026-04-04
> > >
> > > Thank you very much for your thoughtful reviews. Your feedback has significantly improved the quality of our paper. We also greatly enjoyed the rebuttal process and the opportunity to engage with your comments.

---

### Decision · Program_Chairs · 2026-04-30

**Decision:**

Accept (regular)

**Comment:**

The SLAT paper addresses the overthinking problem in large reasoning models by penalizing high probability segments that don't add much value. All four reviewers recommend a weak accept, noting that the method is simple, theoretically grounded, and achieves an impressive 50% length reduction while keeping accuracy competitive. The authors did a good job during the rebuttal by adding experiments on logical reasoning and providing extra ablations on hyperparameters like the probability threshold. While some gaps remain between the theoretical proof and the practical sliding window proxy, the empirical results across various benchmarks are quite strong. Some reviewers pointed out that the evaluation is still mostly focused on math and that position awareness could be a useful future addition. Overall, the consensus is that this is a solid, practical contribution that the community will find useful for making reasoning chains more efficient.